# The nutrient-sensing GCN2 signaling pathway is essential for circadian clock function by regulating histone acetylation under amino acid starvation

Xiao-Lan Liu[1†], Yulin Yang[1,2†], Yue Hu[1], Jingjing Wu[1,2], Chuqiao Han[1,3], Qiaojia Lu[1,2], Xihui Gan[4], Shaohua Qi[5], Jinhu Guo[4], Qun He[5], Yi Liu[6], Xiao Liu[1,2*]

[1]State Key Laboratory of Mycology, Institute of Microbiology, Chinese Academy of Sciences, Beijing, China; [2]College of Life Sciences, University of the Chinese Academy of Sciences, Beijing, China; [3]School of Life Sciences, Yunnan University, Kunming, Yunnan, China; [4]Key Laboratory of Gene Engineering of the Ministry of Education, State Key Laboratory of Biocontrol, School of Life Sciences, Sun Yat-sen University, Guangzhou, China; [5]MOA Key Laboratory of Soil Microbiology, College of Biological Sciences, China Agricultural University, Beijing, China; [6]Department of Physiology, University of Texas Southwestern Medical Center, Dallas, United States

*For correspondence:
liux@im.ac.cn

[†]These authors contributed equally to this work

Competing interest: The authors declare that no competing interests exist.

**Abstract** Circadian clocks are evolved to adapt to the daily environmental changes under different conditions. The ability to maintain circadian clock functions in response to various stresses and perturbations is important for organismal fitness. Here, we show that the nutrient-sensing GCN2 signaling pathway is required for robust circadian clock function under amino acid starvation in *Neurospora*. The deletion of GCN2 pathway components disrupts rhythmic transcription of clock gene *frq* by suppressing WC complex binding at the *frq* promoter due to its reduced histone H3 acetylation levels. Under amino acid starvation, the activation of GCN2 kinase and its downstream transcription factor CPC-1 establish a proper chromatin state at the *frq* promoter by recruiting the histone acetyltransferase GCN-5. The arrhythmic phenotype of the GCN2 kinase mutants under amino acid starvation can be rescued by inhibiting histone deacetylation. Finally, genome-wide transcriptional analysis indicates that the GCN2 signaling pathway maintains robust rhythmic expression of metabolic genes under amino acid starvation. Together, these results uncover an essential role of the GCN2 signaling pathway in maintaining the robust circadian clock function in response to amino acid starvation, and demonstrate the importance of histone acetylation at the *frq* locus in rhythmic gene expression.

## Editor's evaluation

This fundamental work is important in demonstrating that the general amino acid control response to amino acid limitation in Neurospora, which includes the key nutrient-controlled protein kinase Gcn2, is crucial to maintain circadian rhythmic cell growth and transcription of the FRQ gene, the master regulator of rhythmicity. There is an abundance of compelling evidence supporting the conclusions, with rigorous molecular and genetic assays of key mutants impaired for general amino acid control or transcriptional cofactors. The work will be of broad interest to geneticists and molecular biologists, and will be particularly valuable to researchers interested in circadian rhythm or nutrient control of gene expression.

## Introduction

Circadian clocks enable organisms to adapt to the daily environmental changes caused by the earth's rotation (*Bell-Pedersen et al., 2005*; *Dunlap and Loros, 2017*; *Johnson et al., 2017*; *Takahashi, 2017*). Rhythmic gene expression allows different organisms to regulate their daily molecular, cellular, behavioral, and physiological activities. The ability to maintain circadian clock function in response to various stresses and perturbations is an important property of living systems (*Bass, 2012*; *Hogenesch and Ueda, 2011*). Although gene expression is sensitive to temperature changes, temperature compensation is a key feature of circadian clocks to maintain circadian period length at different physiological temperatures (*Hu et al., 2021*; *Narasimamurthy and Virshup, 2021*; *Ode and Ueda, 2018*). DNA damage and translation stress are known to reset the circadian clock through the checkpoint kinase 2 signaling pathway in *Neurospora* and the ATM signaling pathway in mammalian cells (*Diernfellner et al., 2019*; *Oklejewicz et al., 2008*; *Pregueiro et al., 2006*). Cellular redox balance, including oxidative stress, regulates the circadian clock by modulating CLOCK and NPAS2 activity (*Nakahata et al., 2008*; *Rutter et al., 2001*). Nutritional stress, such as a high-fat diet, can disrupt the oscillating metabolites and behavioral rhythms in mice (*Eckel-Mahan et al., 2013*; *Kohsaka et al., 2007*; *Panda, 2016*).

Starvation for all or certain amino acids leads to induced transcription followed by derepression of genes in many amino acid biosynthetic pathways, referred as general amino acid control in yeast and cross-pathway control (CPC) in *Neurospora* (*Hinnebusch, 2005*). General control nonderepressible 2 (GCN2) kinase, called CPC-3 in *Neurospora*, is a serine/threonine kinase that functions as an amino acid sensor (*Battu et al., 2017*; *Efeyan et al., 2015*; *Sattlegger et al., 1998*). GCN2 is activated by accumulated uncharged tRNAs when intracellular amino acids are limited (*Ramirez et al., 1992*; *Wek et al., 1995*). Activated GCN2 phosphorylates the α subunit of eukaryotic initiation factor 2 (eIF2α) to translationally repress protein synthesis (*Lyu et al., 2021*; *Sonenberg and Hinnebusch, 2009*). Meanwhile, it also upregulates the transcription activator GCN4, named CPC-1 in *Neurospora*, which activates amino acid biosynthetic and transport pathways (*Ebbole et al., 1991*; *Hinnebusch, 2005*). Recently, it has been shown that circadian clock control of the GCN2-mediated eIF2α phosphorylation is necessary for rhythmic translation initiation in *Neurospora* (*Ding et al., 2021*; *Karki et al., 2020*). Although amino acid starvation is known to activate the GCN2–GCN4 signaling pathway, how nutrient limitation, especially amino acid starvation, affects circadian clock is not known.

Despite evolutionary divergence in eukaryotes, circadian rhythms are controlled by the conserved transcription/translation-based negative feedback loops (*Bell-Pedersen et al., 2005*). *Neurospora crassa* has been established as one of the best studied model systems for analyzing the molecular mechanism of eukaryotic circadian clocks (*Cha et al., 2015*; *Dunlap and Loros, 2017*; *Heintzen and Liu, 2007*). In the *Neurospora* core circadian negative feedback loop, two PAS-domain-containing transcription factors, White Collar-1 (WC-1) and WC-2 form a complex (WCC) and bind to the C-box of the *frq* promoter to activate its transcription (*Cheng et al., 2001b*; *Crosthwaite et al., 1997*; *Froehlich et al., 2002*). FRQ protein is translated from *frq* mRNA in the cytosol and progressively phosphorylated at about 103 phosphorylation sites, which plays a major role in determining circadian periodicity by regulating the FRQ–CK1 interaction (*Baker et al., 2009*; *Chen et al., 2023*; *Larrondo et al., 2015*; *Liu et al., 2019*; *Liu et al., 2000*; *Tang et al., 2009*). To close the negative feedback loop, FRQ forms a complex with its partner FRQ-interacting RNA helicase (FRH) to inhibit the activity of the WCC by promoting WCC phosphorylation mediated by CK1 and CK2 (*He et al., 2006*; *He and Liu, 2005*; *Schafmeier et al., 2005*; *Wang et al., 2019*).

Chromatin structure and histone modification changes play important roles in regulating the transcription of circadian clock genes (*Papazyan et al., 2016*; *Takahashi, 2017*; *Zhu and Belden, 2020*). In mammals, rhythmic H3K4me3, H3K9ac, and H3K27ac modifications have been shown to be enriched at the promoter of clock genes and positively correlated with gene expression (*Etchegaray et al., 2003*; *Katada and Sassone-Corsi, 2010*; *Koike et al., 2012*). In *Neurospora*, rhythmic binding of WCC to the *frq* promoter is regulated by ATP-dependent chromatin remodeling factors, such as the SWI/SNF complex, the INO80 complex, the chromodomain helicase DNA-binding-1 (CHD1), and the Clock ATPase (CATP) (*Belden et al., 2011*; *Cha et al., 2013*; *Gai et al., 2017*; *Wang et al., 2014*). Histone chaperone FACT complex and histone modification enzymes, SET1, SET2, and RPD3 complex, have also been shown to affect *frq* transcription by regulating rhythmic histone compositions and modifications at the *frq* locus (*Liu et al., 2017*; *Raduwan et al., 2013*; *Sun et al., 2016*). However,

it is still unknown how the chromatin structure is organized to allow rhythmic clock gene expression under nutrient limitation conditions.

In this study, we discovered that the disruption of the GCN2 (CPC-3) signaling pathway abolished robust circadian rhythms under amino acid starvation, which was important for rhythmic expression of metabolic genes. In the GCN2 signaling pathway mutants, amino acid starvation abolished the rhythmic binding of WC-2 at the *frq* promoter by decreasing its histone H3 acetylation levels. Amino acid starvation activated CPC-3 and CPC-1, which re-established a proper chromatin state at the *frq* promoter by recruiting the histone acetyltransferase GCN-5 to allow rhythmic *frq* expression. Furthermore, the inhibition of histone deacetylases could rescue the impaired circadian rhythm phenotypes under amino acid starvation, demonstrating the importance of rhythmic histone acetylation at the *frq* gene promoter for maintaining robust circadian rhythms of gene expression.

## Results
### CPC-3 and CPC-1 are required for robust circadian rhythms under amino acid starvation

To investigate whether the nutrient-sensing GCN2 pathway is involved in regulating circadian clock function under amino acid starvation, we created the *cpc-3* and *cpc-1* knockout mutants (see Materials and methods). In *Neurospora*, *cpc-3* and *cpc-1* encode for the GCN2 and GCN4 homolog, respectively. As expected, the CPC-3-mediated eIF2α phosphorylation and CPC-1 induction by 3-aminotriazole (3-AT) treatment were completely abolished in the *cpc-3^{KO}* strain (*Figure 1—figure supplement 1A*). 3-AT is an inhibitor of the histidine synthesis enzyme encoded by *his-3* which triggers the amino acid starvation response (*Natarajan et al., 2001*). On the other hand, CPC-1 expression and its induction by 3-AT were eliminated in the *cpc-1^{KO}* mutant (*Figure 1—figure supplement 1B*). The circadian conidiation rhythms of the *cpc-3^{KO}* and *cpc-1^{KO}* mutants were examined by race tube assays. Under a normal growth condition (0 mM 3-AT), *cpc-3* deletion had no effect on the circadian period but *cpc-1* deletion led to the period lengthening of ~1.7 hr (*Figure 1A*). The long period of *cpc-1^{KO}* strain could be rescued by the expression of Myc.CPC-1 (*Figure 1—figure supplement 1C*).

To investigate how amino acid starvation affects circadian clock, we treated the wild-type

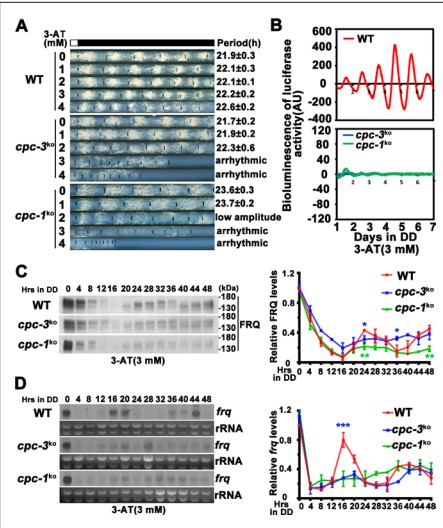

**Figure 1.** CPC-3 and CPC-1 are required for circadian rhythm by regulating rhythmic *frq* transcription in response to amino acid starvation. (**A**) Race tube assay showing that amino acid starvation (3-aminotriazole [3-AT] treatment) disrupted circadian conidiation rhythm of the *cpc-3^{KO}* and *cpc-1^{KO}* strains. 0 mM 3-AT is the normal growth condition. (**B**) Luciferase reporter assay showing that amino acid starvation disrupted rhythmic expression of *frq* promoter-driven luciferase of the *cpc-3^{KO}* and *cpc-1^{KO}* strains. A *frq-luc* transcriptional fusion construct was expressed in *cpc-3^{KO}* and *cpc-1^{KO}* strains grown on the fructose-glucose-sucrose FGS-Vogel's medium with the indicated concentrations of 3-AT, and the luciferase signal was recorded using a LumiCycle in constant darkness (DD) for more than 7 days. Normalized data with the baseline luciferase signals subtracted are shown. (**C**) Western blot showing that amino acid starvation dampened rhythmic expression of FRQ protein of the *cpc-3^{KO}* and *cpc-1^{KO}* strains at the indicated time points in DD ($n = 3$; WT: p = 5.00E−08, *cpc-3^{KO}*: p = 0.0016, *cpc-1^{KO}*: p = 0.0004, RAIN; WT vs *cpc-3^{KO}*: mesor p = 0.1421, amplitude p = 0.0774, phase p = 0.4319; WT vs *cpc-1^{KO}*: mesor p = 0.0614, amplitude p = 0.1920, phase p = 0.4404, CircaCompare). The left panel showing that protein extracts were isolated from WT, *cpc-3^{KO}*, and *cpc-1^{KO}* strains grown in a circadian time course in DD and probed with FRQ antibody. The right panel showing that the densitometric analyses of the results of three independent experiments. (**D**) Northern blot showing that amino acid starvation dampened rhythmic expression of *frq* mRNA of the *cpc-3^{KO}* and *cpc-1^{KO}* strains at the indicated time points in DD ($n = 3$; WT: p = 6.56E−05, *cpc-3^{KO}*: p = 0.0039, *cpc-1^{KO}*: p > 0.05, RAIN; WT vs *cpc-3^{KO}*: mesor p = 0.1153, amplitude p = 0.4316, phase p = 0.0788, CircaCompare). The densitometric analyses of the results from three independent experiments were shown on the right panel. Error bars indicate standard deviation ($n = 3$). *p < 0.05; **p < 0.01; ***p < 0.001; Student's *t* test was used.

*Figure 1 continued on next page*

*Figure 1 continued*

The online version of this article includes the following source data and figure supplement(s) for figure 1:

**Source data 1.** LumiCycle analysis dataset in *Figure 1B*.

**Source data 2.** Scan of western blot probed for FRQ protein and quantification dataset in *Figure 1C*.

**Source data 3.** Scan of Northern blot probed for *frq* mRNA and quantification dataset in *Figure 1D*.

**Figure supplement 1.** CPC-3 and CPC-1 are activated and required for robust circadian conidiation rhythm under histidine starvation.

**Figure supplement 1—source data 1.** Scan of western blot probed for eIF2α protein phosphorylation and CPC-1 protein in *Figure 1—figure supplement 1A*.

**Figure supplement 1—source data 2.** Scan of western blot probed for CPC-1 protein in *Figure 1— figure supplement 1B*.

**Figure supplement 2.** Rhythmic *frq* expression of *cpc-3^KO* and *cpc-1^KO* strain.

**Figure supplement 2—source data 1.** LumiCycle analysis dataset in *Figure 1—figure supplement 2A*.

**Figure supplement 2—source data 2.** LumiCycle analysis dataset in *Figure 1—figure supplement 2B*.

**Figure supplement 2—source data 3.** Scan of western blot probed for FRQ protein and quantification dataset in *Figure 1—figure supplement 2C*.

**Figure supplement 2—source data 4.** Scan of Northern blot probed for *frq* mRNA and quantification dataset in *Figure 1—figure supplement 2D*.

(WT), *cpc-3^KO*, and *cpc-1^KO* strains with different concentrations of 3-AT. As shown in *Figure 1A*, although treatment with 3 or 4 mM 3-AT resulted in modest inhibition of the WT growth rate, the robust circadian conidiation rhythms were maintained. In the *cpc-3^KO* and *cpc-1^KO* strains, however, addition of 3 or 4 mM 3-AT resulted in severe inhibition of growth rates and the loss of circadian conidiation rhythms. To exclude the effect of 3-AT on other target genes, we examined the circadian rhythm of the *his-3^−* strain, which contained a single mutation in the *his-3* gene required for histidine synthesis and could not grow in the medium without histidine. Race tube assays showed that the *his-3^−* strain grew normally and exhibited a robust circadian conidiation rhythm in the presence of histidine ($1.0 \times 10^{-2}$ mg/ml). Although addition of the same amount of histidine could rescue the growth of the *cpc-3^KO his-3^−* strain, it could not rescue its circadian conidiation rhythm (*Figure 1—figure supplement 1D*), indicating that CPC-3 is required for robust circadian rhythms under histidine starvation stress.

To confirm the loss of circadian rhythms at the molecular level, we introduced a *frq* promoter-driven luciferase reporter into the *cpc-3^KO* and *cpc-1^KO* strains. As shown in *Figure 1B* and *Figure 1—figure supplement 2A, B*, the robust circadian rhythms of luciferase activity seen in the WT strain were severely dampened or arrhythmic in the *cpc-3^KO* and *cpc-1^KO* strains upon 3 mM 3-AT treatment. Consistent with these results, western blot analysis showed that, after the initial light/dark transition, rhythms of FRQ levels and its phosphorylation were dampened in the *cpc-3^KO* and *cpc-1^KO* strains in the presence of 3-AT (WT: p = 5.00E−08, *cpc-3^KO*: p = 0.0016, *cpc-1^KO*: p = 0.0004) (*Figure 1C* and *Figure 1—figure supplement 2C*). The statistical tests of circadian rhythms were performed using a circadian statistical analysis tool CircaCompare (*Parsons et al., 2020*) (see Materials and methods). Northern blot analysis showed that the circadian rhythms of *frq* mRNA in the *cpc-3^KO* and *cpc-1^KO* strains were also dampened in the presence of 3-AT (WT: p = 6.56E−05, *cpc-3^KO*: p = 0.0039, *cpc-1^KO*: p > 0.05) and the levels of *frq* mRNA were constantly low in DD (*Figure 1D* and *Figure 1—figure supplement 2D*). Together, these results suggest that the GCN2 pathway is required for a functional clock by regulating rhythmic *frq* transcription in response to amino acid starvation.

## CPC-3 and CPC-1 are required for rhythmic WCC binding in response to amino acid starvation

Since 3-AT treatment resulted in low *frq* mRNA levels in the *cpc-3^KO* and *cpc-1^KO* strains (*Figure 1D*), we first examined the protein levels of WC-1 and WC-2 and found that their levels were higher in the mutants than those in the WT strain at different time points (*Figure 2A, B*). We then performed WC-2 chromatin immunoprecipitation (ChIP) assays to examine whether the WCC binding to the *frq* promoter was affected. As shown in *Figure 2—figure supplement 1A*, WC-2 rhythmically bound to the *frq* C-box in the WT, *cpc-3^KO*, and *cpc-1^KO* strains without 3-AT treatment (WT: p = 2.60E−07, *cpc-3^KO*: p = 0.0016, *cpc-1^KO*: p = 1.06E−05). However, 3-AT treatment resulted in constant low levels of WC-2 binding to the *frq* C-box during a circadian cycle in both *cpc-3^KO* and *cpc-1^KO* strains (*Figure 2C,*

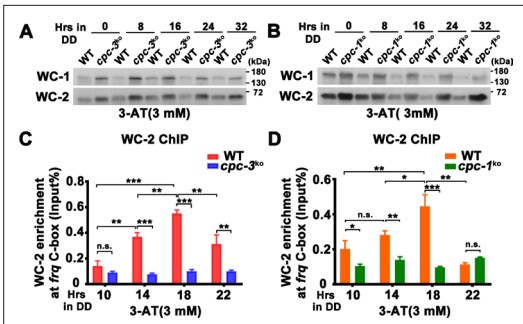

**Figure 2.** CPC-3 and CPC-1 are required for rhythmic WCC binding in response to amino acid starvation. Western blot assay showing that WCC protein levels were elevated in the *cpc-3^KO* (**A**) and *cpc-1^KO* (**B**) strains after 3 mM 3-aminotriazole (3-AT) treatment. Protein extracts were isolated from WT, *cpc-3^KO*, and *cpc-1^KO* strains grown in the indicated time points in DD and probed with WC-1 and WC-2 antibodies. Chromatin immunoprecipitation (ChIP) assay showing that amino acid starvation disrupted rhythmic WC-2 binding at the promoter of *frq* gene in the *cpc-3^KO* (*n* = 3; WT: p = 1.84E−05, *cpc-3^KO*: p > 0.05) (**C**) or *cpc-1^KO* strains (*n* = 3; WT: p = 0.0025, *cpc-1^KO*: p > 0.05) (**D**). Samples were grown for the indicated number of hours in DD prior to harvesting and processing for ChIP using WC-2 antibody. Occupancies were normalized by the ratio of ChIP to Input DNA. Error bars indicate standard deviation (*n* = 3). *p < 0.05; **p < 0.01; ***p < 0.001; Student's *t* test was used.

The online version of this article includes the following source data and figure supplement(s) for figure 2:

**Source data 1.** Scan of western blot probed for WC-1 and WC-2 proteins in WT and *cpc-3^KO* strains with 3-aminotriazole (3-AT) in *Figure 2A*.

**Source data 2.** Scan of western blot probed for WC-1 and WC-2 proteins in WT and *cpc-1^KO* strains with 3-aminotriazole (3-AT) in *Figure 2B*.

**Source data 3.** Chromatin immunoprecipitation (ChIP) analysis dataset in *Figure 2C*.

**Source data 4.** Chromatin immunoprecipitation (ChIP) analysis dataset in *Figure 2D*.

**Figure supplement 1.** WCC binding at the *frq* promoter and WCC phosphorylation levels in the *cpc-3^KO* and *cpc-1^KO* strain.

**Figure supplement 1—source data 1.** Chromatin immunoprecipitation (ChIP) analysis dataset in *Figure 2—figure supplement 1A*.

**Figure supplement 1—source data 2.** Scan of western blot probed for phosphorylation of WC-1 and WC-2 proteins in WT and *cpc-3^KO* strains with 3-aminotriazole (3-AT) in *Figure 2—figure supplement 1B*.

**Figure supplement 1—source data 3.** Scan of western blot probed for phosphorylation of WC-1 and WC-2 proteins in WT and *cpc-1^KO* strains with

*Figure 2 continued on next page*

*Figure 2 continued*

3-aminotriazole (3-AT) in *Figure 2—figure supplement 1C*.

**Figure supplement 1—source data 4.** Chromatin immunoprecipitation (ChIP) analysis dataset in *Figure 2—figure supplement 1D*.

---

**D**). These results indicate that the loss of circadian rhythms in the *cpc-3^KO* and *cpc-1^KO* strains under amino acid starvation is caused by loss of rhythmic *frq* transcription, due to impaired WCC binding at the *frq* promoter.

Because WCC phosphorylation inhibited its transcriptional activation activity (*He et al., 2006*; *He and Liu, 2005*; *Lee et al., 2000*; *Schafmeier et al., 2005*; *Wang et al., 2019*), we also examined WCC phosphorylation profiles and found that 3-AT treatment resulted in hypophosphorylation of WC-1 and WC-2 (which is usually associated with WCC activation) in the *cpc-3^KO* and *cpc-1^KO* strains (*Figure 2—figure supplement 1B, C*). Thus, their reduced WCC binding at the *frq* promoter is not caused by WCC hyperphosphorylation. It should be noted that the overall phosphorylation status of WCC does not always reflect its activity in driving *frq* transcription, which is possibly due to the unknown function of multiple key phosphosites on WCC (*Wang et al., 2019*; *Zhou et al., 2018*).

## CPC-1 is required for the maintenance of chromatin structure in response to amino acid starvation

The low WC-2 binding at the *frq* promoter prompted us to examine the chromatin structure of the *frq* promoter. We first performed ChIP assay to examine the histone and its acetylation levels at the *frq* promoter in the WT strain. Although amino acid starvation had little effect on the histone H2B levels (*Figure 3A*), it led to significantly decreased histone H3 acetylation levels (H3 acetylated at the N-terminus) at the *frq* promoter at high concentrations of 3-AT (*Figure 3B*). These results suggest that amino acid starvation can affect *frq* transcription by reducing the histone acetylation levels at the *frq* promoter.

We then examined whether the CPC-3 and CPC-1 signaling pathway was involved in regulating chromatin structure at the *frq* promoter in response to amino acid starvation. Histone H2B and H3ac ChIP assays at different circadian time points showed that the relative histone H3ac levels were slightly decreased in the *cpc-1^KO*

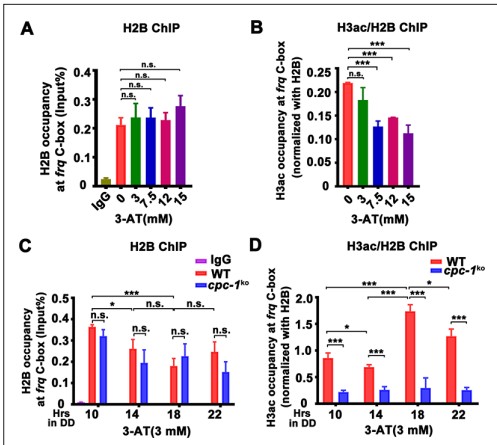

**Figure 3.** CPC-1 is required for the maintenance of chromatin structure in response to amino acid starvation. Chromatin immunoprecipitation (ChIP) assay showing that amino acid starvation slightly increased histone H2B levels (**A**) and dramatically decreased histone H3ac levels (**B**) at the promoter of *frq* gene in the WT strain at DD18 at the indicated concentration of 3-aminotriazole (3-AT). Relative H3ac levels were normalized with H2B levels. (**C, D**) ChIP assay showing that amino acid starvation slightly increased histone H2B levels ($n = 3$; WT: $p = 3.85E−04$, $cpc-1^{KO}$: $p = 0.0364$, RAIN; WT vs $cpc-1^{KO}$: mesor $p = 0.0312$, amplitude $p = 0.2155$, phase $p = 0.2995$, CircaCompare) (**C**) and dramatically decreased histone H3ac levels ($n = 3$; WT: $p = 0.0168$, $cpc-1^{KO}$: $p > 0.05$) (**D**) at the promoter of *frq* gene in the $cpc-1^{KO}$ strain at the indicated time points in DD. Error bars indicate standard deviations ($n = 3$). *$p < 0.05$; ***$p < 0.001$; Student's *t* test was used.

The online version of this article includes the following source data for figure 3:

**Source data 1.** Chromatin immunoprecipitation (ChIP) analysis dataset in *Figure 3A*.

**Source data 2.** Chromatin immunoprecipitation (ChIP) analysis dataset in *Figure 3B*.

**Source data 3.** Chromatin immunoprecipitation (ChIP) analysis dataset in *Figure 3C*.

**Source data 4.** Chromatin immunoprecipitation (ChIP) analysis dataset in *Figure 3D*.

strain compared to the WT strain under normal condition (*Figure 2—figure supplement 1D*). H2B levels were not markedly different between the WT and $cpc-1^{KO}$ strains in the presence of 3 mM 3-AT (*Figure 3C*). However, the relative histone H3ac levels were very different in these two strains in the presence of 3 mM 3-AT: it was rhythmic with a peak at DD18 in the WT strain but was dramatically reduced and arrhythmic in the $cpc-1^{KO}$ strain at different time points in DD (WT: $p = 0.0168$, $cpc-1^{KO}$: $p > 0.05$) (*Figure 3D*). These results indicate that CPC-1 is required for maintaining the proper histone acetylation status at the *frq* promoter under amino acid starvation. The low H3ac levels at the *frq* promoter, which is critical for transcription activation, results in constant low WCC binding and arrhythmic *frq* transcription in the $cpc-1^{KO}$ strain.

## CPC-1 recruits GCN-5 to activate *frq* transcription in response to amino acid starvation

To determine how CPC-1 is involved in regulating histone acetylation levels at the *frq* locus, we examined the occupancy of CPC-1 at the *frq* promoter by ChIP assays using CPC-1 antibody. As shown in *Figure 4A* and *Figure 4—figure supplement 1A*, CPC-1 was found to be rhythmically enriched at the *frq* promoter in the WT strain but not in the $cpc-1^{KO}$ strain under normal (WT: $p = 8.37E−06$, $cpc-1^{KO}$: $p > 0.05$) or amino acid starvation (WT: $p = 7.64E−05$) conditions in DD, peaking at ~DD14, a time point corresponding to the peak of *frq* mRNA levels. Co-immunoprecipitation (Co-IP) assay showed that CPC-1 did not associate with WC-1 or WC-2 (*Figure 4—figure supplement 1B*), suggesting that CPC-1 and WCC bind independently to the *frq* promoter.

How does the GCN2 signaling pathway regulate histone acetylation in response to amino acid starvation? The yeast GCN4 was previously shown to recruit the histone acetyltransferase GCN5 containing (Spt-Ada-Gcn5 acetyltransferase) SAGA complex to selective gene promoters, likely through its physical interaction with the ADA2 subunit (*Barlev et al., 1995*; *Drysdale et al., 1998*; *Kuo et al., 2000*). To test this possibility, we performed Co-IP assay to check the interaction between CPC-1 and the SAGA complex in *Neurospora* in strains expressing the epitope-tagged *Neurospora* SAGA homologs. As shown in *Figure 4B*, Myc-tagged GCN-5 was efficiently immunoprecipitated by the Flag-tagged ADA-2, indicating the existence of an SAGA complex in *Neurospora*. Although the Myc.GCN-5, MYC.CPC-1 or Flag.ADA-2 protein levels were repressed by 3 mM 3-AT treatment (potentially due to global translational inhibition by eIF2α phosphorylation) (*Karki et al., 2020*), the interaction between GCN-5 and ADA-2 was almost the same under either normal or amino acid starved conditions (IP was normalized with Input). Importantly, Myc.CPC-1 was also found to associate specifically with Flag.ADA-2 with/without 3-AT treatment (*Figure 4C*), suggesting that CPC-1 can recruit the SAGA complex to the *frq* promoter

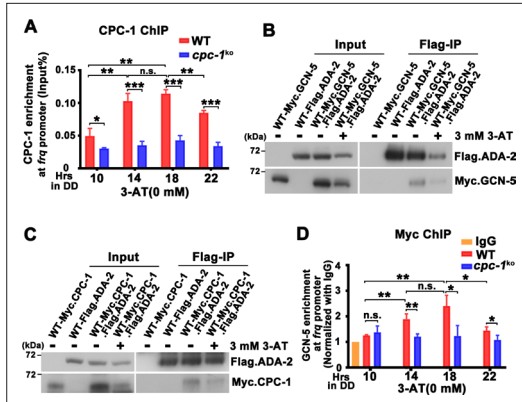

**Figure 4.** CPC-1 recruits GCN-5 to activate *frq* transcription in response to amino acid starvation. (**A**) Chromatin immunoprecipitation (ChIP) assay showing that CPC-1 rhythmically bound at the promoter of *frq* gene (*n* = 3; WT: p = 8.37E−06, *cpc-1^KO*: p > 0.05). WT and *cpc-1^KO* strains grown for the indicated number of hours in DD. Samples were crosslinked with formaldehyde and harvested for ChIP using CPC-1 antibody. CPC-1 ChIP occupancies were normalized by the ratio of ChIP to Input DNA. (**B**) Co-immunoprecipitation (Co-IP) assay showing that Flag.ADA-2 interacted with Myc.GCN-5 with or without 3 mM 3-aminotriazole (3-AT). Flag.ADA-2 and Myc.GCN-5 were co-expressed in the WT strain and immunoprecipitation was performed using Flag antibody. (**C**) Co-IP assay showing that Flag.ADA-2 interacted with Myc.CPC-1 with or without 3 mM 3-AT. Flag.ADA-2 and Myc.CPC-1 were co-expressed in the WT strain and immunoprecipitation was performed using Flag antibody. (**D**) ChIP assay showing that rhythmic GCN-5 binding at the promoter of *frq* gene was dampened in the *cpc-1^KO* strain (*n* = 3; WT: p = 0.0006, *cpc-1^KO*: p > 0.05). Samples were grown for the indicated number of hours in DD prior to harvesting and processing for ChIP as described in (**A**). Error bars indicate standard deviations (*n* = 3). *p < 0.05; **p < 0.01; ***p < 0.001; Student's *t* test was used.

The online version of this article includes the following source data and figure supplement(s) for figure 4:

**Source data 1.** Chromatin immunoprecipitation (ChIP) analysis dataset in *Figure 4A*.

**Source data 2.** Scan of western blot probed for Flag.ADA-2 and Myc.GCN-5 proteins in *Figure 4B*.

**Source data 3.** Scan of western blot probed for Flag.ADA-2 and Myc.CPC-1 proteins in *Figure 4C*.

**Source data 4.** Chromatin immunoprecipitation (ChIP) analysis dataset in *Figure 4D*.

**Figure supplement 1.** CPC-1 rhythmically binds at the *frq* promoter and GCN-5 interacts with WCC.

**Figure supplement 1—source data 1.** Chromatin immunoprecipitation (ChIP) analysis dataset in *Figure 4—figure supplement 1A*.

**Figure supplement 1—source data 2.** Scan of

*Figure 4 continued*

western blot probed for Myc.CPC-1, WC-1, and WC-2 proteins in *Figure 4—figure supplement 1B*.

**Figure supplement 1—source data 3.** Scan of western blot probed for Myc.GCN-5, WC-1, and WC-2 proteins in *Figure 4—figure supplement 1C*.

through its ADA-2 subunit to regulate histone acetylation levels at the *frq* locus under normal or amino acid starvation conditions. Furthermore, immunoprecipitation assays showed that WC-1 and WC-2 also interacted with Myc.GCN-5 (*Figure 4—figure supplement 1C*). These results suggest that CPC-1 can regulate histone acetylation by recruiting the SAGA complex to the *frq* promoter.

To further confirm if CPC-1 can recruit GCN5 to the *frq* promoter, we performed ChIP assay to examine the occupancy of GCN-5 at the *frq* promoter. As shown in *Figure 4D*, Myc-tagged GCN-5 rhythmically bound at the *frq* promoter in DD in the WT strain but its binding was constantly low and arrhythmic in the *cpc-1^KO* strain (WT: p = 0.0006, *cpc-1^KO*: p > 0.05), suggesting that CPC-1 recruits the GCN-5 containing SAGA complex to the *frq* promoter to allow rhythmic histone acetylation levels, which maintain rhythmic WC-2 binding and thus rhythmic *frq* transcription.

## GCN-5 is required for rhythmic H3ac at the *frq* promoter

GCN-5 has been shown to regulate light induction and oxidative stress response in *Neurospora* (*Grimaldi et al., 2006*; *Qi et al., 2018*), but its role in the circadian clock remains unclear. To determine the function of GCN-5 in the circadian clock, we created the *gcn-5* knockout mutant, and found that it exhibited slow growth and lacked a conidiation rhythm (*Figure 5A*). To determine its circadian clock at the molecular level, we introduced the FRQ-LUC reporter (luciferase fused at the C terminus of the FRQ protein) into the *gcn-5^KO* strain (*Larrondo et al., 2015*). As shown in *Figure 5B* and *Figure 5—figure supplement 1B*, a robust circadian rhythm of luciferase activity was observed in the WT strain, but it was quickly dampened after 1 day in DD and became arrhythmic in the *gcn-5^KO* strain. Western blot analysis showed that, after the initial light/dark transition, the rhythmic FRQ abundance and phosphorylation were significantly dampened in the *gcn-5^KO* mutant (WT: p = 5.00E−8, *gcn-5^KO*: p = 0.0016) (*Figure 5C*). Reverse Transcription Quantitative PCR RT-qPCR analysis showed that the circadian rhythms of *frq* mRNA were also

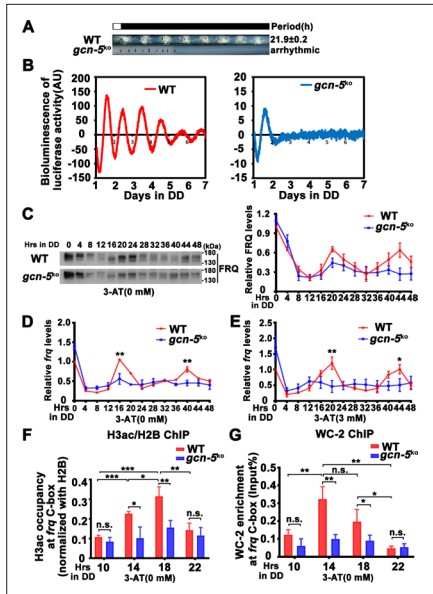

**Figure 5.** GCN-5 is required for rhythmic chromatin structure changes at the *frq* promoter. (**A**) Race tube assay showing that the conidiation rhythm in *gcn-5^KO* strain was lost compared with WT strain. (**B**) Luciferase assay showing that the luciferase activity rhythm was impaired in the *gcn-5^KO* strain after 1 day transition from light to dark. A FRQ-LUC translational fusion construct was expressed in WT and *gcn-5^KO* strains, and the luciferase signal was recorded in DD for more than 7 days. Normalized data with the baseline luciferase signals subtracted are shown. (**C**) Western blot assay showing that rhythmic expression of FRQ protein was dampened in the *gcn-5^KO* strain (n = 3; WT: p = 5.00E−08, *gcn-5^KO*: p = 0.0016, RAIN; WT vs *gcn-5^KO*: mesor p = 0.1421, amplitude p = 0.0774, phase p = 0.4319, CircaCompare). RT-qPCR analysis showing that rhythmic expression of *frq* mRNA was dampened in the *gcn-5^KO* strain without 3-aminotriazole (3-AT; n = 3; WT: p = 8.37E−06, *gcn-5^KO*: p > 0.05) (**D**) or with 3-AT (n = 3; WT: p = 5.39E−12, *gcn-5^KO*: p > 0.05) (**E**). (**F**) Chromatin immunoprecipitation (ChIP) assay showing decreased histone H3ac levels at the promoter of *frq* gene in the *gcn-5^KO* strain at the indicated time points in DD (n = 3; WT: p = 8.10E−05, *gcn-5^KO*: p > 0.05). Relative H3ac levels were normalized with H2B levels. (**G**) ChIP assay showing decreased WC-2 levels at the promoter of *frq* gene in the *gcn-5^KO* strain at the indicated time points in DD (n = 3; WT: p = 0.0003, *gcn-5^KO*: p = 0.0459, RAIN; WT vs *gcn-5^KO*: mesor p = 0.2939, amplitude p = 0.0010, phase p = 0.6933, CircaCompare). Error bars indicate standard deviations (n = 3). *p < 0.05; **p < 0.01; ***p < 0.001; Student's *t* test was used.

The online version of this article includes the following source data and figure supplement(s) for figure 5:

**Source data 1.** LumiCycle analysis dataset in *Figure 5B*.

**Source data 2.** Scan of western blot probed for FRQ protein and quantification dataset in *Figure 5C*.

*Figure 5 continued on next page*

*Figure 5 continued*

**Source data 3.** RT-qPCR analysis dataset in *Figure 5D*.

**Source data 4.** RT-qPCR analysis dataset in *Figure 5E*.

**Source data 5.** Chromatin immunoprecipitation (ChIP) analysis dataset in *Figure 5F*.

**Source data 6.** Chromatin immunoprecipitation (ChIP) analysis dataset in *Figure 5G*.

**Figure supplement 1.** ADA-2 is required for circadian rhythm by regulating rhythmic *frq* expression.

**Figure supplement 1—source data 1.** LumiCycle analysis dataset in *Figure 5—figure supplement 1B*.

**Figure supplement 1—source data 2.** LumiCycle analysis dataset in *Figure 5—figure supplement 1C*.

**Figure supplement 1—source data 3.** Scan of western blot probed for FRQ protein and quantification dataset in *Figure 5—figure supplement 1D*.

**Figure supplement 1—source data 4.** Quantification of Northern blot dataset in *Figure 5—figure supplement 1E*.

**Figure supplement 1—source data 5.** Quantification of Northern blot dataset in *Figure 5—figure supplement 1F*.

severely dampened in the *gcn-5^KO* strain without (WT: p = 8.37E−06, *gcn-5^KO*: p > 0.05) or with 3 mM 3-AT (WT: p = 5.39E−12, *gcn-5^KO*: p > 0.05) (*Figure 5D, E*). Together, these results indicate that GCN-5 is critical for maintaining the function of circadian clock.

ChIP assay results showed that the H3ac levels were significantly decreased at the *frq* promoter, and their rhythmic occupancies were severely dampened in the *gcn-5^KO* strain compared with the WT strain (WT: p = 8.10E−05, *gcn-5^KO*: p > 0.05) (*Figure 5F*). Furthermore, the rhythmic WC-2 binding at the *frq* C-box of the WT strain was dramatically reduced in the *gcn-5^KO* strain in DD (WT: p = 0.0003, *gcn-5^KO*: p = 0.0459) (*Figure 5G*). These results indicate that GCN-5 is critical for circadian clock function by regulating rhythmic chromatin structure changes to allow rhythmic WC-2 binding at the *frq* promoter to drive rhythmic *frq* transcription.

Since ADA-2 interacts with GCN-5 and it is a subunit of the SAGA complex, we also created the *Neurospora ada-2* knockout mutant and examined the function of ADA-2 in the circadian clock. As expected, we found that the circadian phenotypes and *frq* expression of the *ada-2^KO* strain were very similar to those of the *gcn-5^KO* strain (*Figure 5—figure supplement 1A–F*). Together, these results demonstrate the importance of GCN-5 and ADA-2 in the *Neurospora* circadian clock function.

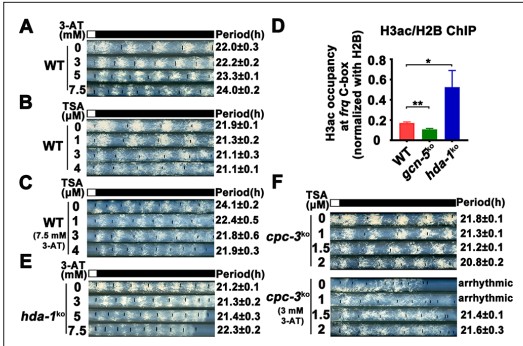

**Figure 6.** Elevated histone acetylation partially rescues impaired circadian rhythm caused by amino acid starvation stress. (**A**) Race tube assay showing that high concentrations of 3-aminotriazole (3-AT) treatment elongated circadian conidiation period of WT strain. (**B**) Race tube assay showing that the high concentrations of Trichostatin A (TSA) treatment shortened circadian conidiation period of WT strain. (**C**) TSA treatment rescued prolonged circadian period of WT strain caused by 3-AT treatment. WT strain was grown on the race tube medium containing 7.5 mM 3-AT and indicated concentrations of TSA in DD. (**D**) Chromatin immunoprecipitation (ChIP) assay showing that H3ac levels were decreased in *gcn-5*[KO] strains and increased in *hda-1*[KO] strains at the promoter of *frq* gene. Error bars indicate standard deviations (*n* = 3). *p < 0.05; **p < 0.01; Student's *t* test was used. (**E**) The *hda-1*[KO] strain exhibited near normal circadian period in the presence of 3-AT. *hda-1*[KO] strains were grown on the race tube medium containing the indicated concentrations of 3-AT in DD. (**F**) TSA treatment rescued the impaired circadian rhythm of *cpc-3*[KO] strain caused by 3-AT treatment. *cpc-3*[KO] strains were grown on the race tube medium containing 3 mM 3-AT and indicated concentrations of TSA in DD.

The online version of this article includes the following source data for figure 6:

**Source data 1.** RT-qPCR analysis dataset in **Figure 6D**.

# Elevated histone acetylation partially rescues circadian clock defects caused by amino acid starvation

Our results above suggest that amino acid starvation results in low histone acetylation levels at the *frq* promoter, which impairs the WC-2 binding and rhythmic *frq* transcription in the *cpc-3*[KO] and *cpc-1*[KO] mutants. To further confirm this conclusion, we hypothesized that the circadian clock defects under amino acid starvation should be rescued by increasing the histone acetylation levels. Trichostatin A (TSA) is a histone deacetylase (HDACs) inhibitor, which can inhibit HDACs activity and increase the histone acetylation levels in *Neurospora* (*Selker, 1998*). We treated the WT strain with different concentrations of 3-AT and TSA, and found that high concentrations of 3-AT lengthened the circadian period of the WT strain to ~24 hr (*Figure 6A*), but high concentrations of TSA resulted in a slight period shortening to 21 hr (*Figure 6B*). When TSA was used together with a high concentration of 3-AT (7.5 mM), the long period phenotype can be gradually rescued by increasing TSA concentrations (*Figure 6C*), which is consistent with our conclusion that the histone acetylation levels changes are responsible for the circadian clock defects caused by amino acid starvation.

GCN-5 is the major histone acetyltransferase responsible for histone acetylation at the *frq* locus in response to amino acid starvation. On the other hand, the histone deacetylase HDA-1 was previously reported as a major histone deacetylase that can antagonize and compete with GCN-5 for recruitment to promoters to deacetylate H3 (*Islam et al., 2011*; *Vogelauer et al., 2000*). ChIP assays showed that H3ac levels were indeed significantly decreased in the *gcn-5*[KO] strain and were markedly increased in the *hda-1*[KO] strain at the *frq* promoter region (*Figure 6D*), indicating that HDA-1 is responsible for histone deacetylation at the *frq* locus. Race tube assays showed that in contrast to period lengthening in the WT strain by 3-AT (*Figure 6A*), the *hda-1*[KO] strain exhibited nearly normal circadian period even in the presence of high 3-AT concentrations (*Figure 6E*), suggesting that reduced histone deacetylation can partially rescue the circadian clock defects in response to amino acid starvation. To further confirm our conclusion, we treated the *cpc-3*[KO] strains with different concentrations of TSA and found that the arrhythmic circadian conidiation rhythm caused by 3-AT treatment could be partially rescued by TSA treatments (*Figure 6F*). Together, these results strongly suggest that the amino acid starvation induced clock defects in the GCN2 signaling pathway mutants are largely due to the decreased histone acetylation at the *frq* promoter, which prevents efficient WC-2 binding and disrupts the rhythmic *frq* transcription.

# Rhythmic expression of CPC-1 activated metabolic genes under amino acid starvation

Circadian clock has been shown to control metabolic processes and rhythmic transcription of metabolic genes (*Baek et al., 2019*; *Hurley et al., 2014*; *Hurley et al., 2018*). To determine the role of the GCN2 pathway in controlling gene expression under amino acid starvation, we performed RNA-seq experiments to analyze the genome-wide mRNA levels in the WT and *cpc-1^{KO}* strains in the presence of 3-AT (12 mM). As shown in *Figure 7A*, 22.1% of genes were found to be upregulated and 11.2% of genes were downregulated in the WT strain after 3-AT treatment compared with normal condition. Specifically, those genes involved in the regulation of oxidoreductase and amino acid metabolism were particularly enriched and were mostly upregulated during the amino acid starvation (*Figure 7B*). However, the differential expression of the 148 upregulated and 127 downregulated genes found in the WT strain was abolished in the *cpc-1^{KO}* strain after 3-AT treatment, suggesting that these genes were regulated by CPC-1 under amino acid starvation (*Figure 7D* and *Figure 7—figure supplement 1A, B*). Similar to those in *Saccharomyces cerevisiae* (*Natarajan et al., 2001*), genes of amino acid biosynthetic pathways, vitamin biosynthetic enzymes, peroxisomal components, and mitochondrial carrier proteins were also identified as CPC-1 targets. Among them, the genes involved in amino acid and vitamin metabolism were particularly enriched and were mostly upregulated during the amino acid starvation (*Figure 7E*). For example, amino acid synthesis genes *his-3* (NCU03139), *arg-1* (NCU02639), *trp-3* (NCU08409), and *ser-2* (NCU01439) were upregulated in the WT strain, but unchanged in the *cpc-1^{KO}* strain under amino acid starvation (*Figure 7A, C*).

To examine whether these CPC-1 activated genes were controlled by circadian clock, we re-analyzed and compared our identified CPC-1 target genes with previously published RNA-seq data of rhythm samples (*Hurley et al., 2014*; *Hurley et al., 2018*). As shown in *Supplementary file 1*, we summarized the rhythmic expression of 79 upregulated and 67 downregulated CPC-1 target genes based on the eJTK Cycle results of *Hurley et al., 2018* (its Supplemental Datasets 1 and 2). We re-analyzed the rhythmicity of CPC-1-targeted genes (148 upregulated and 127 downregulated) using CircaSingle (*Parsons et al., 2020*), and added the p-values in *Supplementary*

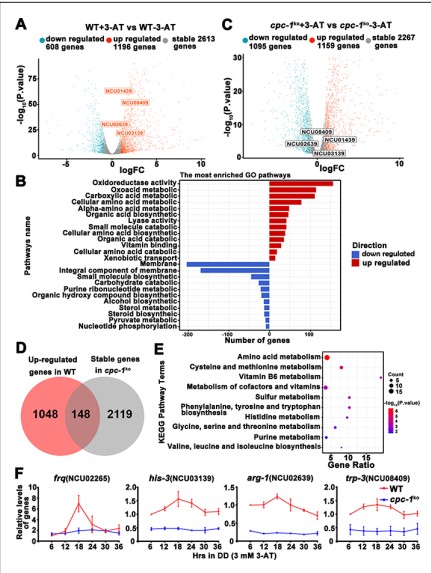

**Figure 7.** Circadian clock control of CPC-1-activated metabolic genes under amino acid starvation. (**A**) Comparison of the transcript expression profiles of the WT strains with and without 12 mM 3-aminotriazole (3-AT) treatment. (**B**) Gene functional enrichment analysis based on the mRNA levels changes for the up- and downregulated genes in the WT strains with and without 12 mM 3-AT treatment. (**C**) Comparison of the transcript expression profiles of the *cpc-1^{KO}* strains with and without 12 mM 3-AT treatment. (**D**) Pie charts showing the overlaps of upregulated genes in the WT strain, but stable genes in the *cpc-1^{KO}* strains after 12 mM 3-AT treatment. (**E**) Gene functional enrichment analysis based on the mRNA levels changes for the overlaps of upregulated genes in the WT strain, but stable genes in the *cpc-1^{KO}* strains after 12 mM 3-AT treatment. (**F**) RT-qPCR analysis showing that amino acid synthetic genes *his-3* (NCU03139) ($n$ = 3; WT: p = 2.21E−05, *cpc-1^{KO}*: p>0.05), *arg-1* (NCU02639) ($n$ = 3; WT: p = 0.0097, *cpc-1^{KO}*: p > 0.05), and *trp-3* (NCU08409) ($n$ = 3; WT: p = 0.0009, *cpc-1^{KO}*: p > 0.05) were activated by CPC-1 and were rhythmic expressed with 3 mM 3-AT treatment. The primers used for RT-qPCR are shown in Key Resources Table.

The online version of this article includes the following source data and figure supplement(s) for figure 7:

**Source data 1.** Raw dataset of RNA-seq analysis in *Figure 7A*.

**Source data 2.** Raw dataset of RNA-seq analysis in *Figure 7C*.

**Source data 3.** RT-qPCR analysis dataset in *Figure 7F*.

**Figure supplement 1.** CPC-1 regulated metabolic genes under amino acid starvation.

**Figure supplement 1—source data 1.** Chromatin immunoprecipitation (ChIP) analysis dataset in *Figure 7—figure supplement 1C*.

**Figure supplement 1—source data 2.** RT-qPCR

*Figure 7 continued on next page*

*Figure 7 continued*

analysis dataset in *Figure 7—figure supplement 1D*.

**Figure supplement 1—source data 3.** RT-qPCR analysis dataset in *Figure 7—figure supplement 1E*.

---

*file 1*. There were 146 rhythmic genes based on the eJTK Cycle, and 132 rhythmic genes based on the CircaSingle. As expected, there were 106 overlapping genes between these two sets of data, confirming the results using eJTK Cycle. Thus, we performed further analysis based on the data from eJTK Cycle. There were about 146/275 (53%) of the CPC-1 up- and downregulated genes under amino acid starvation exhibiting rhythmicity, indicating a highly significant enrichment of CPC-1 regulated genes as clock-controlled genes (p = 3.341905e−06, hypergeometric distribution test). Furthermore, we performed ChIP experiments to examine whether CPC-1 directly activates the expression of amino acid synthetic genes. As shown in *Figure 7—figure supplement 1C*, CPC-1 was found to be constitutively enriched at the promoters of *his-3* (NCU03139), *trp-3* (NCU08409), and *ser-2* (NCU01439) genes (WT (0 mM 3-AT): p > 0.05, WT (3 mM 3-AT): p > 0.05, *cpc-1^{KO}* (0 mM 3-AT): p > 0.05), and the enrichment was enhanced by 3-AT treatment. Because the CPC-1 binding was not rhythmic, we performed RT-qPCR experiments to re-analyze whether those genes were rhythmically transcribed in DD. Consistent with the published RNA-seq analysis results, we found that *frq* and the amino acid synthetic genes such as *his-3* (NCU03139) (WT: p = 2.21E−05, *cpc-1^{KO}*: p > 0.05), *arg-1* (NCU02639) (WT: p = 0.0097, *cpc-1^{KO}*: p > 0.05), and *trp-3* (NCU08409) (WT: p = 0.0009, *cpc-1^{KO}*: p > 0.05) were rhythmically expressed in the WT but not *cpc-1^{KO}* strain under amino acid starvation (*Figure 7F*). In addition, we also found that *ser-2* (NCU01439) (WT: p = 2.25E−05, *cpc-1^{KO}*: p > 0.05) exhibited rhythmic expression in

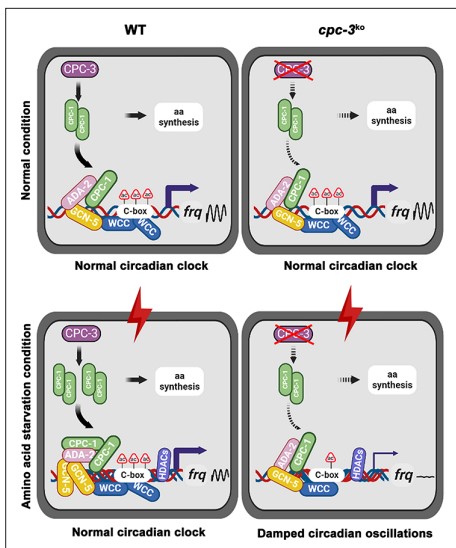

**Figure 8.** Model showing the role of CPC-3 and CPC-1 in maintaining the *Neurospora* circadian rhythm in response to amino acid starvation. Under normal conditions, CPC-1 is expressed at its basal levels in the WT (Left) or *cpc-3^{KO}* (Right) strain, which is required for rhythmic expression of *frq* gene by recruiting the histone acetyltransferase GCN-5 containing SAGA complex to the *frq* promoter. Under amino acid starvation conditions, the chromatin in the *frq* promoter of the WT strain is constitutively compacted (due to decreased H3ac), likely due to activation of histone deacetylases or inhibition of histone acetyltransferases. CPC-3 and CPC-1 signaling pathway was activated by amino acid starvation and the elevated CPC-1 protein would efficiently recruit the histone acetyltransferase GCN-5 containing SAGA complex to promote the histone acetylation levels, which permitted rhythmic WCC binding at the *frq* promoter (Left). Disruption of the CPC-3 and CPC-1 signaling pathway resulted in decreased histone acetylation levels of the *frq* gene promoter, reduced WCC binding and damped circadian oscillations in response to the amino acid starvation stress (Right).

the WT but not in the *cpc-1^{KO}* strain under amino acid starvation (*Figure 7—figure supplement 1D*), even though it was not previously shown to be rhythmic in the published RNA-seq analysis study. These results suggest that many CPC-1-activated metabolic genes under amino acid starvation are regulated by circadian clock. Next, we performed RT-qPCR experiments to detect the mRNA levels of amino acid synthesis genes under normal condition (0 mM 3-AT), and found that *arg-1* (NCU02639) (WT: p = 0.0353), *trp-3* (NCU08409) (WT: p = 0.0436), and *ser-2* (NCU01439) (WT: p = 0.0008) genes, but not the *his-3* (NCU03139) (WT: p > 0.05) gene, were rhythmic in the WT strain in DD (*Figure 7—figure supplement 1E*). Together, these results suggest that the GCN2 signaling pathway functions to maintain the robust circadian clock and rhythmic expression of metabolic genes under amino acid starvation.

## Discussion

Circadian clock drives robust rhythmic gene expression and activities in different environmental and nutritional conditions. Here, we demonstrate that the nutrient-sensing GCN2 pathway plays an unexpected role in maintaining the *Neurospora* circadian clock in response to amino acid starvation stress. Under normal conditions, CPC-1 is expressed at its basal levels in the WT or *cpc-3*$^{KO}$ strain, which is required for rhythmic expression of *frq* gene by recruiting the histone acetyltransferase GCN-5 containing SAGA complex to the *frq* promoter. However, amino acid starvation resulted in compact chromatin structure (due to decreased H3ac) in the *frq* promoter in the WT strain (*Figure 3B*), likely due to activation of the histone deacetylases or inhibition of histone acetyltransferases. The circadian clock-controlled CPC-3 and CPC-1 signaling pathway (*Karki et al., 2020*) is activated by amino acid starvation, and the elevated CPC-1 protein efficiently recruits the histone acetyltransferase GCN-5 containing SAGA complex to the *frq* promoter to increase the histone acetylation levels and loosen the chromatin structure, which permits rhythmic WC-2 binding. Therefore, under amino acid starvation, the disruption of the CPC-3 and CPC-1 signaling pathway results in decreased histone acetylation levels, reduced WC-2 binding at the *frq* promoter and the loss of robust rhythmic *frq* transcription (*Figure 8*).

Maintaining robust rhythmic gene expression and circadian activities in response to various environmental and nutritional stresses is important for the health or survival of different organisms. Circadian clock synchronizes metabolic processes and systemic metabolite levels, while nutrients and energy signals also feedback to circadian clocks to adapt their metabolic state (*Bass, 2012*; *Hurley et al., 2014*; *Klemz et al., 2017*; *Reinke and Asher, 2019*). Amino acid starvation is known to inhibit the global translation efficiency through activating GCN2-mediated eIF2α phosphorylation, which conserves energy and allows cells to reprogram gene expression to relieve stress damage. It was recently shown that circadian clock control of GCN2-mediated eIF2α phosphorylation was necessary for rhythmic translation initiation in *Neurospora* (*Ding et al., 2021*; *Karki et al., 2020*). However, it was previously unknown whether circadian clock would be affected by amino acid starvation stress. After activation of the GCN2-mediated eIF2α phosphorylation by amino acid starvation, a subset of transcripts containing overlapping upstream open reading frames (uORFs) in their 5′ untranslated region (5′ UTR) are efficiently translated, including the yeast transcription factor GCN4, the *Neurospora* CPC-1 and their mammalian ortholog ATF4, which activate the transcription of various amino acid biosynthetic genes (*Hinnebusch, 1984*; *Paluh et al., 1988*; *Vattem and Wek, 2004*). Our ChIP results showed that CPC-1 could rhythmically bind to the region close to C-box at the *frq* promoter to activate *frq* transcription (*Figure 4A*). WCC binding at the *frq* C-box region was slightly decreased under normal condition (*Figure 2—figure supplement 1A*), and dramatically decreased under amino acid starvation in the *cpc-1*$^{KO}$ strain (*Figure 2D*), suggesting that CPC-1 cooperates with WCC to promote *frq* transcription in response to amino acid starvation. Although CPC-1 rhythmically bound at the *frq* promoter, ChIP experiments showed that CPC-1 binding at several selected amino acid biosynthetic genes did not appear to be rhythmic (*Figure 7—figure supplement 1C*). However, our RT-qPCR results (*Figure 7F* and *Figure 7—figure supplement 1*) and the previous RNA-seq data showed that many CPC-1-targeted metabolic genes were rhythmic expressed (*Hurley et al., 2018*). It is possible that the binding of CPC-1 to these promoters is still rhythmic but with low amplitudes. As a result, the limited sensitivity of our ChIP assays failed to detect these rhythms. Alternatively, the rhythmic transcription of these genes might be controlled by the rhythmic transcriptional activation activity of CPC-1 rather than its binding. In addition, the rhythmic binding of WCC or WCC-controlled transcription factors (*Hurley et al., 2014*) might also contribute to their rhythmic transcription.

Amino acid starvation was able to affect gene expression by regulating chromatin modifications. Mammalian transcription factor ATF2 was reported to promote the modification of the chromatin structure in response to amino acid starvation to enhance the transcription of numbers of amino acid-regulated genes (*Bruhat et al., 2007*). Amino acid starvation was also shown to induce reactivation of silenced transgenes and latent HIV-1 provirus by downregulation of histone deacetylase 4 in mammalian cells (*Palmisano et al., 2012*). Here, we found that amino acid starvation suppressed *frq* expression by decreasing the histone acetylation levels likely through activation of histone deacetylases or inhibition of histone acetyltransferases (*Figure 3B*). The activated GCN2 signaling pathway resulted in recruitment of the histone acetyltransferase SAGA complex to the *frq* promoter through its interaction with CPC-1 to re-establish a proper histone acetylation state to relieve the repression (*Figure 3* and

*Figure 4*). Although histone acetylation and deacetylation rhythms have been reported in mammalian cells (*Papazyan et al., 2016*; *Takahashi, 2017*), SAGA complex has also been reported to be involved in circadian regulation through interaction with the CLOCK complex in *Drosophila* (*Mahesh et al., 2020*), their function in circadian clock is unclear in *Neurospora*. Here we demonstrated the important role of GCN-5 in regulating the rhythmic histone acetylation and WC-2 binding at the *frq* promoter (*Figure 5*). Our study unveiled an unsuspected link between nutrient limitation and circadian clock function mediated by the GCN2 signaling pathway. These results provide key insights into the epigenetic regulatory mechanisms of circadian gene expression during amino acid starvation.

The nutrient-sensing GCN2 signaling pathway is highly conserved in eukaryotic cells from yeast to mammals. In the budding yeast *S. cerevisiae* and the filamentous fungus *N. crassa*, the GCN2 kinase responds to nutrient deprivation, whereas it phosphorylates eIF2α and upregulates the master transcription factors GCN4 or CPC-1, respectively, which binds target DNA as a dimer to activate amino acid biosynthetic genes (*Ebbole et al., 1991*; *Hinnebusch, 2005*; *Hope and Struhl, 1987*). In mammalian cells, several GCN2-related kinases can phosphorylate eIF2α in response to various stress conditions, triggering the integrated stress response (*Costa-Mattioli and Walter, 2020*; *Donnelly et al., 2013*). Interestingly, different from the normal circadian period of *cpc-3*[KO] strain in *Neurospora* without amino acid starvation (*Figure 1A*, *Figure 1—figure supplement 1* and *Figure 1—figure supplement 2*), it was reported that GCN2 modulated circadian period by phosphorylation of eIF2α in mammals under normal condition, but it was unknown whether GCN2 was involved in circadian regulation of metabolism under nutrient limitation (*Pathak et al., 2019*). Our results suggest that the GCN2 signaling pathway is required for maintaining circadian clock under amino acid starvation, which is important for robust rhythmic expression of metabolic genes (*Figure 7*). Because GCN2 signaling pathway is important for nutrient sensing, it may also be important for nutritional compensation (*Kelliher et al., 2023*) and plays a role in maintaining the robustness of rhythms in a range of nutritional conditions. Time-restricted feeding prevents obesity and metabolic syndrome through circadian-related mechanisms (*Chaix et al., 2019*), but how eating pattern affects circadian regulation is unclear. The conservation of the GCN2 signaling pathway and our results here suggest that GCN2 may play an important role in mediating circadian regulation of metabolism during nutrient limitation caused by feeding restrictions in mammals. Together, our studies suggest a conserved role of the GCN2 signaling pathway in maintaining the robustness of circadian clock under nutrient starvation in eukaryotes.

## Materials and methods

**Key resources table**

| Reagent type (species) or resource | Designation | Source or reference | Identifiers | Additional information |
|---|---|---|---|---|
| Strain, strain background (*Neurospora crassa*) | 87-3 (*ras-1*[bd], a) | Dr.Yi Liu's Laboratory | | |
| Strain, strain background (*Neurospora crassa*) | 301-6 (*ras-1*[bd], *his-3*-, A) | Dr.Yi Liu's Laboratory | | |
| Strain, strain background (*Neurospora crassa*) | *ras-1*[bd];*cpc-3*[KO] | Fungal Genetics Stock Center | NCU01187 | |
| Strain, strain background (*Neurospora crassa*) | *ras-1*[bd];*cpc-1*[KO] | Fungal Genetics Stock Center | NCU04050 | |
| Strain, strain background (*Neurospora crassa*) | *ras-1*[bd];*gcn-5*[KO] | Fungal Genetics Stock Center | NCU10847 | |
| Strain, strain background (*Neurospora crassa*) | *ras-1*[bd];*ada-2*[KO] | Fungal Genetics Stock Center | NCU04459 | |

*Continued on next page*

*Continued*

| Reagent type (species) or resource | Designation | Source or reference | Identifiers | Additional information |
|---|---|---|---|---|
| Strain, strain background (*Neurospora crassa*) | *ras-1^{bd};hda-1^{KO}* | Fungal Genetics Stock Center | NCU01525 | |
| Strain, strain background (*Neurospora crassa*) | *ras-1^{bd};cpc-1^{KO}*, cpc-1-Myc.CPC-1 | Dr.Xiao Liu's Laboratory | | |
| Strain, strain background (*Neurospora crassa*) | 301-6, cfp-Myc.CPC-1, cfp-Flag.ADA-2 | Dr.Xiao Liu's Laboratory | | |
| Strain, strain background (*Neurospora crassa*) | 301-6, cfp-Myc. GCN-5, cfp-Flag. ADA-2 | Dr.Xiao Liu's Laboratory | | |
| Strain, strain background (*Neurospora crassa*) | *frq-luc* | Dr.Yi Liu's Laboratory | | |
| Strain, strain background (*Neurospora crassa*) | *ras-1^{bd};cpc-3^{KO}, frq-luc* | Dr.Xiao Liu's Laboratory | | |
| Strain, strain background (*Neurospora crassa*) | *ras-1^{bd};cpc-1^{KO}, frq-luc* | Dr.Xiao Liu's Laboratory | | |
| Strain, strain background (*Neurospora crassa*) | FRQ-LUC | Dr. Luis Larrondo's Laboratory | | |
| Strain, strain background (*Neurospora crassa*) | *ras-1^{bd};gcn-5^{KO}, FRQ-LUC* | Dr.Xiao Liu's Laboratory | | |
| Strain, strain background (*Neurospora crassa*) | *ras-1^{bd};ada-2^{KO}, FRQ-LUC* | Dr.Xiao Liu's Laboratory | | |
| Antibody | Rabbit polyclonal anti-Histone H2B | Abcam | Cat# ab1790 | 1:3000 |
| Antibody | Rabbit polyclonal anti-Histone H3ac | Millipore | Cat# 06-599 | 1:3000 |
| Antibody | Mouse monoclonal anti-c-Myc | TransGen | Cat# HT101 | 1:3000 |
| Antibody | Mouse monoclonal anti-Flag | Sigma | Cat# F1804 | 1:3000 |
| Antibody | Rabbit polyclonal anti-P-eIF2α | Abcam | Cat# ab32157 | 1:3000 |
| Antibody | Rabbit polyclonal anti-CPC-1 | Dr.Xiao Liu's Laboratory | | 1:3000 |
| Antibody | Rabbit polyclonal anti-FRQ | Dr.Yi Liu's Laboratory | | 1:3000 |
| Antibody | Rabbit polyclonal anti-WC-1 | Dr.Yi Liu's Laboratory | | 1:4000 |
| Antibody | Rabbit polyclonal anti-WC-2 | Dr.Yi Liu's Laboratory | | 1:8000 |
| Sequence-based reagent | *frq-F* | Dr.Xiao Liu's Laboratory | RT-qPCR | GCAGTGTCATTGACGACTTG |

*Continued*

| Reagent type (species) or resource | Designation | Source or reference | Identifiers | Additional information |
|---|---|---|---|---|
| Sequence-based reagent | *frq-R* | Dr.Xiao Liu's Laboratory | RT-qPCR | CCTCCAACTCACGTTTCTTTC |
| Sequence-based reagent | *his-3-F* | Dr.Xiao Liu's Laboratory | RT-qPCR | CCTCGTTCGTCAAGCACATTA |
| Sequence-based reagent | *his-3-R* | Dr.Xiao Liu's Laboratory | RT-qPCR | CTCCTCAACCTTAGCCAACTG |
| Sequence-based reagent | *trp-3-F* | Dr.Xiao Liu's Laboratory | RT-qPCR | ACCTATATCCTTCAGAACCAATACG |
| Sequence-based reagent | *trp-3-R* | Dr.Xiao Liu's Laboratory | RT-qPCR | GCTCGGTATCCTTCCAGTTG |
| Sequence-based reagent | *ser-2-F* | Dr.Xiao Liu's Laboratory | RT-qPCR | GCTGCTAACGGTGACTACTT |
| Sequence-based reagent | *ser-2-R* | Dr.Xiao Liu's Laboratory | RT-qPCR | GGTGAGGATGATGTTGTTGAG |
| Sequence-based reagent | *arg-1-F* | Dr.Xiao Liu's Laboratory | RT-qPCR | CCCATCATTGCCCGTGCCC |
| Sequence-based reagent | *arg-1-R* | Dr.Xiao Liu's Laboratory | RT-qPCR | TGACGACCCTGGAAGCGAG |
| Sequence-based reagent | *β-tubulin-F* | Dr.Xiao Liu's Laboratory | RT-qPCR | GCGTATCGGCGAGCAGTT |
| Sequence-based reagent | *β-tubulin-R* | Dr.Xiao Liu's Laboratory | RT-qPCR | CCTCACCAGTGTACCAATGCA |
| Sequence-based reagent | *frq* C-box-F | Dr.Xiao Liu's Laboratory | ChIP-qPCR | GTCAAGCTCGTACCCACATC |
| Sequence-based reagent | *frq* C-box-R | Dr.Xiao Liu's Laboratory | ChIP-qPCR | CCGAAAGTATCTTGAGCCTCC |
| Sequence-based reagent | *frq* promoter-F | Dr.Xiao Liu's Laboratory | ChIP-qPCR | GTTGCCGTGACTCCCCCTTG |
| Sequence-based reagent | *frq* promoter-R | Dr.Xiao Liu's Laboratory | ChIP-qPCR | CCGAAAGTATCTTGAGCCTCC |
| Sequence-based reagent | *his-3* ChIP-F | Dr.Xiao Liu's Laboratory | ChIP-qPCR | TTTTCATAAAGCCCGAGTCT |
| Sequence-based reagent | *his-3* ChIP-R | Dr.Xiao Liu's Laboratory | ChIP-qPCR | CAGGTATTGTGCTGTTCCCC |
| Sequence-based reagent | *trp-3* ChIP-F | Dr.Xiao Liu's Laboratory | ChIP-qPCR | AATCGGGTGAGTCAAAGGCG |
| Sequence-based reagent | *trp-3* ChIP-R | Dr.Xiao Liu's Laboratory | ChIP-qPCR | CGAGCAAGAGGGAGAGGTGT |
| Sequence-based reagent | *ser-2* ChIP-F | Dr.Xiao Liu's Laboratory | ChIP-qPCR | GGGACAAAAGCAGTGATTCTA |
| Sequence-based reagent | *ser-2* ChIP-R | Dr.Xiao Liu's Laboratory | ChIP-qPCR | CGATTTACATCCATCTGAGA |
| Sequence-based reagent | *frq* northern-F | Dr.Xiao Liu's Laboratory | Northern blot | TAATACGACTCACTATAGGGCCTTCGTTGGATATCCATCATG |
| Sequence-based reagent | *frq* northern-R | Dr.Xiao Liu's Laboratory | Northern blot | GAATTCTTGCAGGGAAGCCGG |
| Software, algorithm | ImageJ | https://imagej.nih.gov/ij/ | | |

*Continued on next page*

*Continued*

| Reagent type (species) or resource | Designation | Source or reference | Identifiers | Additional information |
| --- | --- | --- | --- | --- |
| Software, algorithm | LumiCycle | https://actimetrics.com/products/lumicycle/ | | |
| Software, algorithm | CircaCompare | https://github.com/RWParsons/circacompare/; *Parsons et al., 2020* | | |

## Strains, culture conditions, and race tube assay

The 87-3 (*ras-1^{bd}*, a) and 301-6 (*ras-1^{bd}*, *his-3⁻*, A) strain was used as the wild-type (WT) strain in this study. Knockout mutants were all generated based on the *ras-1^{bd}* background (*Belden et al., 2007*). *cpc-3^{KO}* (NCU01187), *cpc-1^{KO}* (NCU04050), *gcn-5^{KO}* (NCU10847), *ada-2^{KO}* (NCU04459), and *hda-1^{KO}* (NCU01525) strains were obtained from the Fungal Genetic Stock Center (FGSC) and were crossed with a *ras-1^{bd}* strain to create the *ras-1^{bd};cpc-3^{KO}*, *ras-1^{bd};cpc-1^{KO}*, *ras-1^{bd};gcn-5^{KO}*, *ras-1^{bd};ada-2^{KO}*, and *ras-1^{bd};hda-1^{KO}* strains (*Colot et al., 2006*). Constructs with the *cpc-1* promoter driving expression of Myc.CPC-1 were introduced into the *cpc-1^{KO}* strains at the *his-3* (NCU03139) locus by homologous recombination. Constructs with the *cfp* promoter driving expression of Flag.ADA-2 were introduced into the 301-6, cfp-Myc.CPC-1 or 301-6, cfp-Myc.GCN-5 strains by random insertion with nourseothricin selection (*He et al., 2020*). Positive transformants were identified by western blot analyses, and homokaryon strains were isolated by microconidia purification with 5 μm filters.

Liquid cultures were grown in minimal media (1× Vogel's, 2% glucose). For rhythmic experiments, *Neurospora* was cultured in petri dishes in liquid medium for 2 days. The *Neurospora* mats were cut into discs and transferred into medium-containing flasks and were harvested at the indicated time points.

The medium for race tube assay contained 1× Vogel's salts, 0.1% glucose, 0.17% arginine, 50 ng/ml biotin, and 1.5% agar. After entrainment of 24 hr in the constant light condition, race tubes were transferred to constant darkness conditions and marked every 24 hr. The circadian period of the *Neurospora* strain could be calculated according to the ratio between the distance of marked conidia band positions and the distance of conidiation bands.

## Luciferase reporter assay

The luciferase reporter assay was performed as reported previously (*Gooch et al., 2008*; *Larrondo et al., 2015*; *Liu et al., 2017*). The luciferase reporter construct (*frq-luc*) containing the luciferase gene under the control of the *frq* promoter, was introduced into 301-6, *cpc-3^{KO}* or *cpc-1^{KO}* strains by transformation. The luciferase reporter construct (FRQ-LUC) containing a luciferase fused to the C terminus of the FRQ protein, was introduced into *gcn-5^{KO}* or *ada-2^{KO}* strains by crossing. Firefly luciferin (final concentration of 50 μM) was added to autoclaved FGS-Vogel's medium containing 1× FGS (0.05% fructose, 0.05% glucose, 2% sorbose), 1× Vogel's medium, 50 μg/l biotin, and 1.8% agar. Conidia suspension was placed on autoclaved FGS-Vogel's medium and grown in constant light overnight. The cultures were then transferred to constant darkness, and luminescence was recorded in real time using a LumiCycle after 1 day in DD. The data were then normalized with LumiCycle analysis software by subtracting the baseline luciferase signal, which increases as cell grows.

## Protein analysis

Protein extraction, quantification, and western blot analyses were performed as previously described (*Liu et al., 2017*). Briefly, tissues were ground in liquid nitrogen with a mortar and pestle and suspended in ice-cold extraction buffer (50 mM Hydroxyethylpiperazine Ethane Sulfonic Acid HEPES (pH 7.4), 137 mM NaCl, 10% glycerol) with protease inhibitors (1 μg/ml Pepstatin A, 1 μg/ml Leupeptin, and 1 mM phenylmethylsulfonyl fluoride PMSF). After centrifugation, protein concentrations were measured using protein assay dye (Bio-Rad). For western blot analyses, equal amounts of total protein (40 μg) were loaded in each protein lane of 7.5% or 10% sodium dodecyl sulfate–polyacrylamide gel electrophoresis (SDS–PAGE) gels containing a ratio of 37.5:1 acrylamide/bisacrylamide. After electrophoresis, proteins were transferred onto PVDF membranes, and western blot analyses using FRQ,

WC-1, WC-2, P-eIF2α (Abcam, ab32157), and CPC-1 antibodies were performed. Western blot signals were detected by X-ray films and scanned for quantification.

To detect the phosphorylation levels of WC-1 and WC-2, PPase inhibitors (25 mM NaF, 10 mM Na$_4$P$_2$O$_7$·10H$_2$O, 2 mM Na$_3$VO$_4$, 1 mM ethylenediaminetetraacetic acid EDTA) were made fresh and added to the protein extraction buffer. Proteins were loaded in each protein lane of 7.5% SDS–PAGE gels containing a ratio of 149:1 acrylamide/bisacrylamide.

## RNA analysis

RNA was extracted with Trizol and further purified with 2.5 M LiCl as described previously (*Liu et al., 2017*). For Northern blot analysis, equal amounts of total RNA (20 μg) were loaded onto agarose gels. After electrophoresis, the RNA was transferred onto nitrocellulose membrane. The membrane was probed with [$^{32}$P] UTP (PerkinElmer)-labeled RNA probes specific for *frq*. RNA probes were transcribed in vitro from PCR products by T7 RNA polymerase (Invitrogen, AM1314M) with the manufacturer's protocol. The *frq* primers used for the template amplification are shown in Key Resources Table.

For RT-qPCR, the cultures of WT and *cpc-1$^{KO}$* strains were collected at the indicated time points in constant darkness in liquid growth medium (1× Vogel's, 2% glucose). RT-qPCR were performed as previously described (*Cui et al., 2020*). Each RNA sample (1 μg) was subjected to reverse transcription with HiScript II reverse transcriptase (Vazyme, R223), and then amplified by real-time PCR (Bio-Rad, CFX96). For RT-qPCR, primers target the coding genes of *frq* (NCU02265), *his-3* (NCU03139), *ser-2* (NCU01439), *trp-3* (NCU08409), *his-4* (NCU06974), *arg-1* (NCU02639), and *arg-10* (NCU08162) were designed, and the β-tubulin (NCU04054) was used as an internal control. The primers used for RT-qPCR are shown in Key Resources Table.

## Generation of antiserum against CPC-1

Two CPC-1 peptides (SELDLLDFATFDGG and RDKPLPPIIVEDPS) were synthesized and used as the antigens to generate rabbit polyclonal antisera (ABclonal) as described previously (*Cui et al., 2020*; *Zhou et al., 2013*).

## Co-IP analysis

Immunoprecipitation analyses were performed as previously described (*Cao et al., 2018*; *Cheng et al., 2001b*). Briefly, *Neurospora* proteins were extracted as described above. For each immunoprecipitation reaction, 2 mg protein and 2 μl c-Myc (TransGen, HT101), 2 μl Flag (Sigma, F1804), or 2 μl WC-2 antibody (*Cheng et al., 2001a*) were used. After incubation with antibody for 3 hr, 40 μl GammaBind G Sepharose beads (GE Healthcare, 17061801) were added, and samples were incubated for 1 hr. Immunoprecipitated proteins were washed three times using extraction buffer before western blot analysis. IP experiments were performed using cultures harvested in constant light.

## ChIP analysis

ChIP assays were performed as previously described (*Cui et al., 2020*; *Zhou et al., 2013*) with 1 mg protein used for each immunoprecipitation reaction. The ChIP reaction was carried out with 2 μl WC-2 (*Cheng et al., 2001a*), CPC-1, H2B (Abcam, ab1790), or H3ac (Millipore, 06-599) antibody. Immunoprecipitated DNA was quantified by real-time qPCR. Occupancies were normalized by the ratio of ChIP to Input DNA. ChIP was performed using 2 μl c-Myc monoclonal antibody (TransGen, HT101) to examine occupancy of Myc.GCN-5. Occupancies of ChIP were normalized using IgG. The primers used for ChIP-qPCR are shown in Key Resources Table.

## RNA-seq analysis

The WT and *cpc-1$^{KO}$* strains were cultured with or without 12 mM 3-AT treatment in constant light. Total RNAs were extracted using Trizol reagents. Libraries were prepared according to the manufacturer's instructions and analyzed using 150 bp paired-end Illumina sequencing (Annoroad Gene Technology, Beijing). After sequencing, the raw data were treated and mapped to the genome of *Neurospora crassa* and transformed into expression value. The gene expression levels were scored by fragments per kb per million (FPKM) method. The differences in gene expression between samples was compared by comparing FPKM values, and those with fold change more than 1 (False Discovery Rate FDR <0.1) were thought to be differentially expressed genes. The functional category enrichments,

including Gene Ontology (GO) and Kyoto Encyclopedia of Genes and Genomes KEGG terms, were analyzed. The KEGG pathway enrichment was evaluated based on hypergeometric distribution, and the R package 'ggplot2' version 3.3.6 was used to visualize the enrichment results.

## Quantification and statistical analysis

Quantification of western blot data were performed using Image J software. Error bars are standard deviations for ChIP assays from at least three independent technical experiments, and standard error of means for race tube assays from at least three independent biological experiments, unless otherwise indicated. Statistical significance was determined by Student's *t* test. Statistical tests for the presence of rhythmicity and differences between two rhythms in parameters, including amplitude, phase, and mesor (the midline estimating statistic of rhythms) were analyzed using CircaSingle and CircaCompare (https://rwparsons.shinyapps.io/circacompare/) (*Liu et al., 2021*; *Parsons et al., 2020*). CircaCompare was used to compare the differences between two rhythmic datasets, and CircaSingle was used to re-analyze the rhythmic parameters of the RNA-seq results by eJTK Cycle. Amplitude refers to half of the difference between the peak and trough of a given response variable, phase refers to the time at which the response variable peaks, and mesor refers to the rhythm-adjusted mean level of a response variable around which a wave function oscillates. Statistical tests for the presence of rhythmicity and statistically significant differences between two groups of rhythmic parameters are indicated by p-values. A p-value <0.05 indicates the presence of rhythmicity, but a p-value >0.05 indicates the loss of rhythmicity. The results of the CircaSingle statistical tests are added in *Supplementary file 1*. The results of the CircaCompare statistical tests are summarized in *Supplementary file 2*.

## Data and materials availability

All data generated or analyzed during this study are included in the manuscript and supporting files. Our generated RNA sequencing data have been deposited in GEO under accession code GSE220169. RNA sequencing data from *Hurley et al., 2014* were previously deposited to the NCBI SRA under accession number SRP046458 (*Hurley et al., 2014*). Materials are available from the corresponding authors upon reasonable request.

## Acknowledgements

We thank Dr. Luis Larrondo for providing the translational fused FRQ-LUC plasmid and strain, and Dr. Shaojie Li for providing *Neurospora* mutant strains. We thank Dr. Jay Dunlap and Dr. Jennifer Hurley for providing *Neurospora* RNA-seq information. We also thank Dr. Linqi Wang, Dr. Wenbing Yin, and members of our lab for assistance and discussions. This work was supported by grants from National Natural Science Foundation of China (32170092, 31970079), National Key Research and Development Program of China (2021YFA0911300), Strategic Priority Research Program of the Chinese Academy of Sciences (XDA28030402), Beijing Natural Science Foundation (5202020), and CAS Interdisciplinary Innovation Team to Xiao Liu; National Natural Science Foundation of China (32200056) to Xiaolan Liu; National Key Research and Development Program of China (2018YFA0900500) and National Natural Science Foundation of China (32170560) to Qun He; National Institutes of Health (R35 GM118118) and the Welch Foundation (I-1560) to Yi Liu. The funders had no role in study design, data collection and analysis, decision to publish, or preparation of the manuscript.

## Additional information

### Funding

| Funder | Grant reference number | Author |
|---|---|---|
| National Natural Science Foundation of China | 32170092 | Xiao Liu |
| National Natural Science Foundation of China | 31970079 | Xiao Liu |

| Funder | Grant reference number | Author |
|---|---|---|
| National Key Research and Development Program of China | 2021YFA0911300 | Xiao Liu |
| Strategic Priority Research Program of the Chinese Academy of Sciences | XDA28030402 | Xiao Liu |
| Beijing Natural Science Foundation | 5202020 | Xiao Liu |
| CAS Interdisciplinary Innovation Team | | Xiao Liu |
| National Natural Science Foundation of China | 32200056 | Xiao-Lan Liu |
| National Key Research and Development Program of China | 2018YFA0900500 | Qun He |
| National Natural Science Foundation of China | 32170560 | Qun He |
| National Institutes of Health | R35 GM118118 | Yi Liu |
| Welch Foundation | I-1560 | Yi Liu |

The funders had no role in study design, data collection, and interpretation, or the decision to submit the work for publication.

## Author contributions

Xiao-Lan Liu, Conceptualization, Formal analysis, Funding acquisition, Validation, Investigation, Writing - original draft, Writing - review and editing; Yulin Yang, Conceptualization, Formal analysis, Validation, Investigation; Yue Hu, Jingjing Wu, Chuqiao Han, Qiaojia Lu, Formal analysis, Validation, Investigation; Xihui Gan, Shaohua Qi, Investigation; Jinhu Guo, Resources, Formal analysis; Qun He, Resources, Formal analysis, Funding acquisition; Yi Liu, Resources, Formal analysis, Funding acquisition, Writing - review and editing; Xiao Liu, Conceptualization, Resources, Formal analysis, Supervision, Funding acquisition, Writing - original draft, Writing - review and editing

## Author ORCIDs

Xiao-Lan Liu ⓘ http://orcid.org/0000-0002-1755-3387
Yi Liu ⓘ http://orcid.org/0000-0002-8801-9317
Xiao Liu ⓘ http://orcid.org/0000-0001-6053-132X

## Decision letter and Author response

Decision letter https://doi.org/10.7554/eLife.85241.sa1
Author response https://doi.org/10.7554/eLife.85241.sa2

## Additional files

### Supplementary files

- MDAR checklist
- Supplementary file 1. Analyses of CPC-1 regulated genes.
- Supplementary file 2. Analyses of rhythmic datasets by CircaCompare.

### Data availability

All data generated or analyzed during this study are included in the manuscript and supporting files. Our generated RNA sequencing data have been deposited in GEO under accession code GSE220169.

The following dataset was generated:

| Author(s) | Year | Dataset title | Dataset URL | Database and Identifier |
|---|---|---|---|---|
| Liu XL, Yang Y, Liu X | 2022 | The nutrient-sensing GCN2 signaling pathway is essential for circadian clock function by regulating histone acetylation under amino acid starvation | https://www.ncbi.nlm.nih.gov/geo/query/acc.cgi?acc=GSE220169 | NCBI Gene Expression Omnibus, GSE220169 |

The following previously published dataset was used:

| Author(s) | Year | Dataset title | Dataset URL | Database and Identifier |
|---|---|---|---|---|
| Hurley JM, Dasgupta A, Dunlap JC | 2014 | Analysis of clock-regulated genes in Neurospora reveals widespread posttranscriptional control of metabolic potential | https://www.ncbi.nlm.nih.gov/bioproject/PRJNA250475 | NCBI BioProject, PRJNA250475 |

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
