## [Editor Report]

This fundamental work is important in demonstrating that the general amino acid control response to amino acid limitation in Neurospora, which includes the key nutrient-controlled protein kinase Gcn2, is crucial to maintain circadian rhythmic cell growth and transcription of the FRQ gene, the master regulator of rhythmicity. There is an abundance of compelling evidence supporting the conclusions, with rigorous molecular and genetic assays of key mutants impaired for general amino acid control or transcriptional cofactors. The work will be of broad interest to geneticists and molecular biologists, and will be particularly valuable to researchers interested in circadian rhythm or nutrient control of gene expression.

---

## [Decision Letter]

**Decision letter after peer review:**

Thank you for submitting your article "The nutrient-sensing GCN2 signaling pathway is essential for circadian clock function by regulating histone acetylation under amino acid starvation" for consideration by *eLife*. Your article has been reviewed by 3 peer reviewers, one of whom is a member of our Board of Reviewing Editors, and the evaluation has been overseen by Kevin Struhl as the Senior Editor. The following individuals involved in the review of your submission have agreed to reveal their identity: Joanna Chiu (Reviewer #2); Bin Wang and Jay C Dunlap (Reviewer #3, co-reviewers).

The reviewers have discussed their reviews with one another, and the Reviewing Editor Alan Hinnebusch has drafted this to help you prepare a revised submission.

Essential revisions:

1) Conduct statistical analysis to assess the significance of the circadian rhythms claimed for FRQ mRNA abundance, and for WC complex binding, H3Ac levels, GCN-5 binding, and CPC-1 binding at the FRQ promoter.

2) The data are not compelling that CPC-1 occupancy at FRQ is rhythmic. If the statistically significant rhythmic binding of CPC-1 is not demonstrated, then the molecular basis of the rhythmic recruitment of Gcn5 and H3Ac will be unclear. In that event, the authors may need to consider the model that CPC-1 assists constitutively in recruiting Gcn5 and attendant H3 acetylation, and that some other factor is involved in producing the circadian rhythms of these events, and in turn, the rhythmic binding of WC complex at the FRQ promoter.

3) It appears that the experiments to examine the involvement of GCN-5 and ADA-2 in the rhythmic transcription of FRQ were performed only in non-starvation conditions. If this was indeed the case, then at a minimum, key conclusions would have to be qualified. More likely, the experiments would have to be extended to include starvation conditions.

4) Provide clarification about why knock out of GCN5 reduces FRQ protein expression and diminishes its rhythm without corresponding effects on FRQ mRNA abundance (Figure 5C vs 5D). Are the data lacking in precision or accuracy, or is a more complex model required to explain these results?

5) The data provided to support the claim of rhythmic transcription of the four candidate amino acid biosynthetic genes in Figure 7F is not very compelling. If the statistical analysis fails to confirm their rhythmic transcription in WT, it might be possible to bolster their claim by evaluating published RNA-seq data (sampled every 2 hrs over 2 days, done in triplicate) for known CPC-1 target genes. Because these published experiments were not carried out in amino acid starvation conditions however, the measured expression of the biosynthetic gene transcripts might be dominated by transcription factors that bind constitutively rather than by CPC-1. In the absence of published data supporting rhythmic transcription of amino acid biosynthetic genes regulated by CPC-1, the authors should consider conducting additional RNA-Seq experiments at different time points in amino acid-starved cells. Alternatively, it might be possible to provide RT-qPCR data on additional candidate genes, coupled with statistical analysis of rhythmic transcription. Failing that, the authors would likely need to soften or eliminate claims of rhythmic transcription of these genes.

6) The proposed rhythmic transcription of amino acid biosynthetic genes would also predict rhythmic binding by CPC-1 in their promoters under starvation conditions. Conducting ChIP analysis of CPC-1 binding at a few candidate biosynthetic genes that show convincing rhythmic transcription would be highly desirable, and if observed, might compensate for less compelling transcript data. Such measurements would also be relevant to the question noted above of whether CPC-1 binding at the FRQ promoter itself is truly rhythmic.

7) It is necessary to add new text to address queries raised by Referees #2 and #3 about various findings that require additional clarification or qualification.

8) It is important to provide raw source data for all measurements.

*Reviewer #1 (Recommendations for the authors):*

i) Explain why knock out of GCN5 eliminates rhythmic FRQ expression but does not have a corresponding effect on FRQ mRNA abundance.

ii) Determine whether elimination of HDA1 restores H3ac and WCC binding at FRQ, and rhythmic growth, in 3AT-treated cpc-1 and gcn5 mutants.

iii) Provide stronger evidence for rhythmic transcription of aa biosynthetic genes.

iv) Possibly attempt to determine whether CPC-1 occupancy at aa biosynthetic genes (and at FRQ itself) is rhythmic.

*Reviewer #2 (Recommendations for the authors):*

1. The authors should use circadian statistics (e.g. CircaCompare; Parsons et al. 2020) to compute the phase and amplitude of the mRNA, DNA binding of WC complex, and H3Ac rhythms. This will allow them to compare between rhythms and provide statistical significance values, rather than just providing qualitative descriptions, e.g. "slightly decreased" in lines 189-190. This will be valuable when comparing rhythms between strains and between nutrient conditions.

2. Line 119: The authors indicate that the rhythmicity of histone acetylation at the frq gene promoter is lost in many of the mutants they tested, e.g. CPC-3 and CPC-1. However, it is not clear whether these mutants simply have a lower level of H3ac (Figure 3D), resulting in a lower level of WC binding to frq promoter (Figure 2C-D), and consequently lower level of frq mRNA (Figure 1D). When the levels of these measures are low, it is difficult to observe any rhythms. Furthermore, is it really necessary to emphasize the maintenance of rhythms? When clock mRNA levels are too low, clocks are normally disrupted anyway.

3. The authors propose that GCN5 works downstream of CPC-1 and CPC-3. If that is the case, why does frq mRNA in gcn-5 KO strain show dampened but consistently high level (Figure 5D), rather than a consistently low level as in the case of cpc-1 and cpc-3 ko strains? Can the authors comment on this? Also, it appears that H3ac and WC binding to frq promoter is lower compared to WT (Figure 5E-F). Can the authors explain why frq mRNA level is higher given H3ac and WC binding to frq promoter is lower?

4. There is no data in this study showing that the GCN2 pathway is activated in amino acid-starved conditions, only that it is required to maintain robust frq and conidiation rhythms. Can the authors clarify how they are defining "activation of the GCN2 pathway" in this study? For example, is it recruitment of GCN-5 and SAGA complex to frq promoter?

5. The experiments to examine the involvement of GCN-5 and ADA-2 were performed in normal conditions (no amino acid starvation). Unlike cpc-1 and cpc-3 KO strains, gcn-5 and ada-2 KO strains showed severely disrupted frq rhythms, suggesting they are normally required for robust circadian rhythms. If GCN-5 and the SAGA complex are normally involved in regulating H3ac rhythms in the frq loci, how does the GCN2 pathway modulates the activity of GCN-5 and SAGA complex in conditions of amino acid starvation? Are the interactions between GCN2/4 with GCN-5 and SAGA complex different in normal vs amino acid starved conditions? The authors should clarify their model.

6. Given that the GCN2 pathway is important for nutrient sensing, the authors should not disregard the alternative hypothesis that the GCN2 pathway may be important for nutrient compensation and plays a role in maintaining the robustness of rhythms in a range of nutrient conditions.

[Editors' note: further revisions were suggested prior to acceptance, as described below.]

Thank you for resubmitting your work entitled "The nutrient-sensing GCN2 signaling pathway is essential for circadian clock function by regulating histone acetylation under amino acid starvation" for further consideration by *eLife*. Your revised article has been evaluated by Kevin Struhl (Senior Editor) and a Reviewing Editor, Alan Hinnebusch.

It appears that you have satisfied some, but not all, of the major issues raised about the previous version of the paper. It was recognized that you provided new evidence indicating that knockout of GCN5 eliminates rhythmic FRQ expression with a corresponding effect on FRQ mRNA abundance. While you also provided new ChIP evidence suggesting that CPC-1 occupancy at FRQ is rhythmic; it was provided only to the referees. You performed statistical analysis of rhythmicity using the CircaCompare tool; however, the application of tests and the resulting statistics have not been presented in a transparent or comprehensive manner. It also appears that you incorrectly cited published evidence regarding the rhythmicity of amino acid biosynthetic gene expression. These and a few other issues are fully described below:

1. The description of the CircaCompare statistical tests of rhythmicity is inadequate and needs to be better explained, providing a brief "layman's" description in the Methods of the meaning of the statistical parameters and how the tests were applied to their data. In addition, the results need to be documented completely for all experiments in which rhythmicity was concluded to exist for any parameter. While the authors added time-series statistics in certain figure legends, they did not indicate the p-values for rhythmicity, and the asterisks in figure panels seem to refer to t-tests rather than to RAIN/CircaCompare statistics. To increase transparency, the results of the CircaCompare statistical tests could be summarized in an additional supplementary table.

2. Discrepancies remain regarding whether transcription of amino acid (AA) biosynthetic genes is rhythmic as, contrary to the statement in the Response to Reviewers, the publication by Hurley et al. 2014 does not show ser-2 (NCU01439), trp-3 (NCU08409) or arg-1 (NCU02639) to be rhythmic. The authors should note that the original data were re-analyzed using better statistical tools, described in Suppl Dataset S1 of Hurley et al. (2018), and in the re-analyzed data, ser-2(NCU01439) is not rhythmic at the RNA level, although it is rhythmic at the protein level; trp-3(NCU08409) is rhythmic at the RNA level, and arg-1 (NCU02639)- is rhythmic at the RNA level. The authors need to carefully examine the results of Hurley et al. (2018) and provide an accurate summary of the presence or absence of rhythmicity in the expression of all known CPC-1-induced target genes, possibly with an additional supplementary table that lists the key statistical criteria for the assessments. They should also explain to the reviewers why they incorrectly cited the findings of Hurley et al. (2014) in the revised manuscript.

3. The new ChIP data for CPC-1 binding at the AA genes should be included as a supplementary figure with the addition of the appropriate explanatory text in Results and Discussion, including the suggestion made in their Response to Reviewers of rhythmic binding of WCC or WCC-controlled transcription factors, at the AA genes and whether any evidence exists in the literature for the latter.

4. Clarify whether the authors are interpreting their data to indicate that GCN5/SAGA perform the same functions at FRQ promoter in both non-starved and starved cells and are simply being additionally mobilized by CPC-1 in starved cells, possibly by altering the model in Figure 8.

*Reviewer #1 (Recommendations for the authors):*

The authors appear to have satisfied some, but not all, of the major issues raised about the previous version of the paper. They have provided new evidence indicating that knock out of GCN5 eliminates rhythmic FRQ expression with a corresponding effect on FRQ mRNA abundance. They also provided new ChIP evidence suggesting that CPC-1 occupancy at FRQ is rhythmic. However, the following issues seem to require additional attention and adequate responses from the authors.

1. To support the conclusions of rhythmic FRQ mRNA abundance, WCC complex binding, H3Ac levels, GCN-5 binding, and CPC-1 binding at the FRQ promoter, and of expression levels of amino acid biosynthetic genes in WT cells, they claim to have conducted a statistical analysis using CircaCompare. However, I could not find even a cursory explanation of this tool and the meaning of the different parameters it evaluates in Methods, nor a description of how it was being applied in comparing results for WT and mutants. In addition, it is generally unclear what data have been analyzed with CircaCompare and what the outcomes were. In particular, the legends often state that data were judged to be arhythmic in a cpc mutant without indicating whether it was significantly rhythmic in WT. What is needed is a table that lists clearly all the CircaCompare tests that were conducted and the P-values for all of the parameters that were analyzed, with a final column indicating whether the data are rhythmic or arhythmic and from what results this assessment was made. In short, for each experiment, it is necessary to provide justification using CircaCompare that the data are significantly rhythmic in WT, not only that they are arhythmic in mutants, and to make the CircaCompare tests and results completely transparent, accessible, and understandable to a general audience.

2. The aforementioned shortcoming is particularly notable regarding the expression of amino acid (AA) biosynthetic genes in WT cells. It is unclear whether the CircaCompare analysis actually confirms rhythmic mRNA expression for these genes in WT cells, both for the group of genes they examined by RT-PCR and those interrogated using published RNA-seq data. It is also unclear why they are showing the published RNA-seq results for only 4 genes out of a presumably much larger number of AA biosynthetic genes known to be induced by CPC-1 in starved cells. Did many other AA genes not show rhythmic transcription? It would useful to have a table summarizing the CircaCompare analysis of the RNA-seq data for all known CPC-1 target genes.

3. This last issue of whether AA biosynthetic gene transcription is truly rhythmic was exacerbated by their new CPC-1 ChIP data (presented only for referees) indicating constitutive CPC-1 occupancy at all four AA biosynthetic genes they tested, in contrast to its apparently rhythmic binding at the FRQ gene. They suggest that the AA biosynthetic genes might exhibit rhythmic binding of WCC, or some other transcriptional activator; however, is there any evidence that WCC binds to these AA genes, or that any other activator besides WCC binds to promoters rhythmically? I remain skeptical of the claim of rhythmic transcription of AA biosynthetic genes that are induced by CPC-1 in starved cells.

*Reviewer #2 (Recommendations for the authors):*

The authors have adequately addressed my comments and concerns from the previous manuscript version, in some cases with additional experiments. I greatly appreciate their effort. I think this study is a nice contribution to the field and will inform on the metabolic regulation of circadian rhythms at the chromatin level. One minor comment that will help clarify this complex model for readers.

In the previous version, I commented:

"The experiments to examine the involvement of GCN-5 and ADA-2 were performed in normal conditions (no amino acid starvation). Unlike cpc-1 and cpc-3 KO strains, gcn-5 and ada-2 KO strains showed severely disrupted frq rhythms in normal nutrient conditions, suggesting they are normally already required for robust circadian rhythms. If GCN-5 and the SAGA complex are normally involved in regulating H3ac rhythms in the frq loci, how does GCN2 pathway modulates the activity of GCN-5 and SAGA complex in conditions of amino acid starvation? Are the interactions between GCN2/4 with GCN-5 and SAGA complex different in normal vs amino acid starved conditions? "

Based on the additional experiments and their text edit, there appears to be no difference in the activity of GCN-5 and SAGA complex in normal vs amino acid-starved conditions. It's doing what it normally does in normal conditions even when it is in amino acid-starved conditions. Its activity is however more necessary in amino acid-starved conditions because the chromatin at frq promoter is constitutively compacted in amino acid-starved conditions. So since CPC-3 and CPC-1 are necessary to activate the GCN-5/ADA-2 complex and since an amino acid-starved condition activates the GCN2 pathway and induced CPC-1 expression, the arrhythmic phenotype in circadian rhythm is more severe in CPC-3 and CPC-1 KO mutants in amino acid starved condition when compared to normal condition.

I wonder if the model figure (Figure 8) would be more clear to readers if it includes the normal scenario instead of just having the amino acid-starved conditions. Besides that, I think the manuscript is much approved.

*Reviewer #3 (Recommendations for the authors):*

There are frequent and numerous issues with English usage that should be corrected. There are too many of these to mark each one, and they begin with problems in the Abstract.

Here are a few examples just from the Introduction:

"Circadian clocks are evolved to adapt to the daily environmental changes caused by the earth rotation". Should be earth's rotation "The ability to maintain circadian clock.

54 function in response to various stress and perturbations is an important property of living systems.

55 (Bass, 2012; Hogenesch and Ueda, 2011). Although gene expression is sensitive to temperature.

56 changes, temperature compensation is a key feature of circadian clocks that maintain circadian.

57 period length".

Should read "in response to various stresses" and "compensation is a key feature of circadian clocks that maintains".

Lines 128 and 136, the same sentence appears twice: "3-AT is an inhibitor…".

Lines 168, 170, 171, and 177, cpc-1 KO to cpc-1KO; line 364, "related echanisms(Chaix,"

to "related mechanisms (Chaix,"; line 381, "nourseothricin selection(L." to "nourseothricin selection (L.". Line 460, "using 150bp" to "using 150 bp".

Line 174 could add the citation to Wang et al., 2019 here as was done in citing the same conclusion on lines 95/96.

Line 230 should be "we crossed ras^-1^[bd] to the gcn-5KO strain obtained from the FGSC" or something to that effect. Likewise in the Materials and methods and elsewhere, "bd" should be ras^-1^[bd] or ras^-1^bd. Likewise on line 247 with ada-2 where ras^-1^[bd] was crossed into the knockout strain.

More generally it is important that strains always be referred to by their correct and full genotypes unless a clear method of abbreviations is stated early in the main text and invariably applied throughout. This is not trivial, especially with chromatin-modifying proteins. For instance, Belden (PLoS Gen 2011) found a synthetic lethality between ∆chd1 and ras^-1^[bd].

In the model (Figure 8), formation of the CPC-1 oligomer was not confirmed experimentally, and this could be mentioned.

[Editors' note: further revisions were suggested prior to acceptance, as described below.]

Thank you for resubmitting your work entitled "The nutrient-sensing GCN2 signaling pathway is essential for circadian clock function by regulating histone acetylation under amino acid starvation" for further consideration by *eLife*. Your revised article has been evaluated by Kevin Struhl (Senior Editor) and a Reviewing Editor.

The manuscript has been improved but there are some remaining issues that need to be addressed, as outlined below:

1. In the new Supplementary file S1. The authors have provided lists of 148 cpc-1 up-regulated genes and 127 cpc-1 down-regulated genes, of which 79 up-regulated genes apparently show rhythmic expression and 67 down-regulated genes apparently show rhythmic expression. They have not however provided the statistics of the CircaCompare tests of rhythmicity for the WT strain, which needs to be added for each gene listed in the new file S1. In addition, they should provide new text in the RESULTS explaining that ~53% of both the up- and down-regulated genes exhibit rhythmicity and indicate that this represents a highly significant enrichment of all cpc-1 regulated genes as judged by the hypergeometric distribution test, providing the P-value. The file S1 should also include a notes sheet fully explaining the data provided there.

2. A notes sheet should also be added to the new Supplementary file S2 in which the results in each sheet should be linked to data in the corresponding figures, and also stipulating the difference between results listed in different sheets for FRQ versus frq (Protein vs. mRNA?). In short, each Supplementary file should be understandable as a stand-alone document.

3. There are also some grammatical errors in the revised text that should be corrected.

---

## [Author Response]

Essential Revisions:1) Conduct statistical analysis to assess the significance of the circadian rhythms claimed for FRQ mRNA abundance, and for WC complex binding, H3Ac levels, GCN-5 binding, and CPC-1 binding at the FRQ promoter.

We revised as suggested by reviewer 2 using CircaCompare.

2) The data are not compelling that CPC-1 occupancy at FRQ is rhythmic. If the statistically significant rhythmic binding of CPC-1 is not demonstrated, then the molecular basis of the rhythmic recruitment of Gcn5 and H3Ac will be unclear. In that event, the authors may need to consider the model that CPC-1 assists constitutively in recruiting Gcn5 and attendant H3 acetylation, and that some other factor is involved in producing the circadian rhythms of these events, and in turn, the rhythmic binding of WC complex at the FRQ promoter.

To address this concern, we now performed ChIP assays using the CPC-1 antibody instead of Myc antibody. As shown in the revised Figure 4A, the CPC-1 enrichment at DD14/DD18 was significantly higher than that of DD10/DD22. In addition, we also analyzed the circadian rhythm of CPC-1 binding at the *frq* promoter with CircaCompare, which showed that the CPC-1 occupancy at the *frq* promoter was rhythmic.

3) It appears that the experiments to examine the involvement of GCN-5 and ADA-2 in the rhythmic transcription of FRQ were performed only in non-starvation conditions. If this was indeed the case, then at a minimum, key conclusions would have to be qualified. More likely, the experiments would have to be extended to include starvation conditions.

Thanks for the suggestion. We have now examined the rhythmic transcription of *frq* in both amino acid non-starvation and starvation conditions in the *gcn-5^KO^* (Figure 5D and 5E) and *ada-2^KO^* mutants (Figure 5—figure supplement 1E and 1F). The results were found to be similar with/without amino acid starvation in the *gcn-5^KO^* and *ada-2^KO^* mutants.

4) Provide clarification about why knock out of GCN5 reduces FRQ protein expression and diminishes its rhythm without corresponding effects on FRQ mRNA abundance (Figure 5C vs 5D). Are the data lacking in precision or accuracy, or is a more complex model required to explain these results?

To address this concern, we performed RT-qPCR analysis to compare the rhythms of *frq* mRNA in the WT and *gcn-5^KO^* mutant. The results showed that the *frq* mRNA were decreased in the *gcn-5^KO^* mutant with or without 3-AT, especially at DD16h and DD40h, when *frq* mRNA peaked in WT (Figure 5D and 5E). Thus, the *frq* mRNA and FRQ protein results are consistent with each other.

5) The data provided to support the claim of rhythmic transcription of the four candidate amino acid biosynthetic genes in Figure 7F is not very compelling. If the statistical analysis fails to confirm their rhythmic transcription in WT, it might be possible to bolster their claim by evaluating published RNA-seq data (sampled every 2 hrs over 2 days, done in triplicate) for known CPC-1 target genes. Because these published experiments were not carried out in amino acid starvation conditions however, the measured expression of the biosynthetic gene transcripts might be dominated by transcription factors that bind constitutively rather than by CPC-1. In the absence of published data supporting rhythmic transcription of amino acid biosynthetic genes regulated by CPC-1, the authors should consider conducting additional RNA-Seq experiments at different time points in amino acid-starved cells. Alternatively, it might be possible to provide RT-qPCR data on additional candidate genes, coupled with statistical analysis of rhythmic transcription. Failing that, the authors would likely need to soften or eliminate claims of rhythmic transcription of these genes.

The rhythmic expression of these candidate amino acid biosynthetic genes are supported by the following evidence. First, we analyzed the published rhythmic RNA-seq data from Dr. Jay Dunlap’s lab (Hurley JM *et al.*, 2014, PMID: 25362047) and found that these amino acid biosynthetic gene (*ser-2*, *trp-3*, *arg-1*) were expressed rhythmically. Our analysis data is now presented in Figure 7—figure supplement 1C. Second, we also performed RT-qPCR analysis to examine the relative mRNA levels of *his-3*, *ser-2*, *trp-3*, and three additional amino acid biosynthetic genes (*his-4*, *arg-1* and *arg-10)* at different time points (sampled every 6 hrs over 30 hrs). We found that these amino acid biosynthetic genes were rhythmic expressed in the WT but not in the *cpc-1*^KO^ strain with/without 3-AT treatment (Figure 7 F and Figure 7—figure supplement 1D and 1E).

6) The proposed rhythmic transcription of amino acid biosynthetic genes would also predict rhythmic binding by CPC-1 in their promoters under starvation conditions. Conducting ChIP analysis of CPC-1 binding at a few candidate biosynthetic genes that show convincing rhythmic transcription would be highly desirable, and if observed, might compensate for less compelling transcript data. Such measurements would also be relevant to the question noted above of whether CPC-1 binding at the FRQ promoter itself is truly rhythmic.

To address this concern, we performed ChIP using our CPC-1 antibody to analyze CPC-1 occupancy at the *frq* promoter and a few amino acid biosynthetic genes including *his-3*, *ser-2* and *trp-3*. As the results shown in Figure 4A, CPC-1 binding at the *frq* promoter was rhythmic (measured with CircaCompare). However, the CPC-1 occupancy at the *his-3*, *ser-2* and *trp-3* promoters was not rhythmic (measured with CircaCompare). These results suggested that CPC-1 binding is rhythmic at the *frq* promoter but not at the amino acid biosynthetic genes. Thus, the rhythmic transcription of amino acid biosynthetic genes might be controlled by rhythmic binding of WCC or WCC-controlled transcription factors but not by rhythmic CPC-1 binding.

**Author response image 1. sa2fig1:** 

7) It is necessary to add new text to address queries raised by Referees #2 and #3 about various findings that require additional clarification or qualification.

Revised as suggested. We have edited the text and revised the figures in the revised manuscript.

8) It is important to provide raw source data for all measurements.

Revised as suggested. Raw source data of measurements are now shown in the revised folder of source data.

Reviewer #1 (Recommendations for the authors):i) Explain why knock out of GCN5 eliminates rhythmic FRQ expression but does not have a corresponding effect on FRQ mRNA abundance.

Please see response to the Essential revisions point 4.

ii) Determine whether elimination of HDA1 restores H3ac and WCC binding at FRQ, and rhythmic growth, in 3AT-treated cpc-1 and gcn5 mutants.

Because TSA treatment partially rescued the rhythmicity of *cpc-3^KO^* strain after 3-AT treatment (Figure 6F), we created *hda-1^KO^cpc-3^KO^* double mutant and performed race tube assay with addition of different concentrations of 3-AT. The results showed that the *hda-1^KO^cpc-3^KO^* mutant was sensitive to 3-AT, becoming arhythmic with 2 or 3 mM 3-AT, which is similar to *cpc-3^KO^* strain (data shown in Author response image 2). These results suggested that in addition to HDA-1, other HDACs are also involved in restoring H3ac and WCC binding at the *frq* promoter in the *cpc-3 ^KO^*, *cpc-1 ^KO^* or *gcn-5 ^KO^* mutants under amino acid starvation conditions. We are planning to further investigate this mechanism in the future.

iii) Provide stronger evidence for rhythmic transcription of aa biosynthetic genes.

Please see response to Essential revisions point 5.

iv) Possibly attempt to determine whether CPC-1 occupancy at aa biosynthetic genes (and at FRQ itself) is rhythmic.

Please see response to Essential revisions point 6.

Reviewer #2 (Recommendations for the authors):1. The authors should use circadian statistics (e.g. CircaCompare; Parsons et al. 2020) to compute the phase and amplitude of the mRNA, DNA binding of WC complex, and H3Ac rhythms. This will allow them to compare between rhythms and provide statistical significance values, rather than just providing qualitative descriptions, e.g. "slightly decreased" in lines 189-190. This will be valuable when comparing rhythms between strains and between nutrient conditions.

As suggested, we used CircaCompare to analyze our data.

2. Line 119: The authors indicate that the rhythmicity of histone acetylation at the frq gene promoter is lost in many of the mutants they tested, e.g. CPC-3 and CPC-1. However, it is not clear whether these mutants simply have a lower level of H3ac (Figure 3D), resulting in a lower level of WC binding to frq promoter (Figure 2C-D), and consequently lower level of frq mRNA (Figure 1D). When the levels of these measures are low, it is difficult to observe any rhythms. Furthermore, is it really necessary to emphasize the maintenance of rhythms? When clock mRNA levels are too low, clocks are normally disrupted anyway.

Thanks for the suggestions. We re-analyzed the data using CircaCompare and revised the manuscript.

3. The authors propose that GCN5 works downstream of CPC-1 and CPC-3. If that is the case, why does frq mRNA in gcn-5 KO strain show dampened but consistently high level (Figure 5D), rather than a consistently low level as in the case of cpc-1 and cpc-3 ko strains? Can the authors comment on this? Also, it appears that H3ac and WC binding to frq promoter is lower compared to WT (Figure 5E-F). Can the authors explain why frq mRNA level is higher given H3ac and WC binding to frq promoter is lower?

Please see response to Essential revisions point 4.

4. There is no data in this study showing that the GCN2 pathway is activated in amino acid-starved conditions, only that it is required to maintain robust frq and conidiation rhythms. Can the authors clarify how they are defining "activation of the GCN2 pathway" in this study? For example, is it recruitment of GCN-5 and SAGA complex to frq promoter?

Thanks for the question. CPC-3, the GCN2 homolog in *Neurospora*, is the only eIF2α kinase responsible for eIF2α phosphorylation at serine 51(Karki S et al. 2020, PMID: 32355000). As shown in the revised Figure 1—figure supplement 1A, the eIF2α phosphorylation and CPC-1 were induced by 3-AT treatment in the WT but not in the *cpc-3*^KO^ strain. These results demonstrate that the GCN2 pathway is activated by amino acid starvation, and as a result, the CPC-1 expression is activated to recruit the SAGA complex to the *frq* promoter.

5. The experiments to examine the involvement of GCN-5 and ADA-2 were performed in normal conditions (no amino acid starvation). Unlike cpc-1 and cpc-3 KO strains, gcn-5 and ada-2 KO strains showed severely disrupted frq rhythms, suggesting they are normally required for robust circadian rhythms. If GCN-5 and the SAGA complex are normally involved in regulating H3ac rhythms in the frq loci, how does the GCN2 pathway modulates the activity of GCN-5 and SAGA complex in conditions of amino acid starvation? Are the interactions between GCN2/4 with GCN-5 and SAGA complex different in normal vs amino acid starved conditions? The authors should clarify their model.

As mentioned above, our data suggested that GCN-5 and ADA-2 are required for robust circadian rhythms under normal conditions. As suggested, we did detect dampened rhythmic expression of *frq* in the *gcn-5^KO^* and *ada-2^KO^* strains under amino acid starvation (Figure 5D and 5E and Figure 5—figure supplement 1E and 1F). We also performed Co-IP to compare the difference of interactions between CPC-1 with ADA-2 and GCN5 with ADA-2 under normal and amino acid starved conditions. The results showed that although the Myc.GCN-5, MYC.CPC-1 or Flag.ADA-2 protein level was repressed by 3 mM 3-AT treatment (likely due to global translational inhibition by induced eIF2α phosphorylation) (Karki S et al. 2020, PMID: 32355000), the interactions between CPC-1 with ADA-2 and GCN-5 with ADA-2 were almost the same under normal and amino acid starved conditions (IP was normalized with Input) (Figure 4B and 4C). These results indicated that amino acid starved conditions had little impact on the protein interactions between CPC-1 with GCN-5 and SAGA complex.

In our model, we proposed that amino acid starvation resulted in compact chromatin structure (due to decreased H3ac) in the *frq* promoter in the WT strain (Figure 3B), likely due to activation of histone deacetylases or inhibition of histone acetyltransferases. Amino acid starvation activates GCN2 pathway and induces CPC-1 expression. The induced CPC-1 can recruit GCN5-containing SAGA complex to the *frq* promoter to loosen the chromatin structure, promoting *frq* rhythmic transcription under starvation conditions. However, in the *cpc-3^KO^* mutants, CPC-1 could not effectively recruit GCN5 containing SAGA complex to *frq* promoter, resulting in arrhythmic *frq* transcription. We have now clarified our model in the revised discussion.

6. Given that the GCN2 pathway is important for nutrient sensing, the authors should not disregard the alternative hypothesis that the GCN2 pathway may be important for nutrient compensation and plays a role in maintaining the robustness of rhythms in a range of nutrient conditions.

Thanks for the suggestion. We now discussed the alternative hypothesis in the revised manuscript. “Because GCN2 signaling pathway is important for nutrient sensing, it may be important for nutrient compensation and plays a role in maintaining the robustness of rhythms in a range of nutrient conditions”.

Reviewer #3 (Recommendations for the authors):There are frequent and numerous issues with English usage that should be corrected. There are too many of these to mark each one, and they begin with problems in the Abstract.Here are a few examples just from the Introduction:"Circadian clocks are evolved to adapt to the daily environmental changes caused by the earth rotation". Should be earth's rotation "The ability to maintain circadian clock.54 function in response to various stress and perturbations is an important property of living systems.55 (Bass, 2012; Hogenesch & Ueda, 2011). Although gene expression is sensitive to temperature.56 changes, temperature compensation is a key feature of circadian clocks that maintain circadian.57 period length".Should read "in response to various stresses" and "compensation is a key feature of circadian clocks that maintains".

Thanks for the suggestions. The manuscript is revised as suggested.

Lines 128 and 136, the same sentence appears twice: "3-AT is an inhibitor…".Lines 168, 170, 171, and 177, cpc-1 KO to cpc-1KO; line 364, "related echanisms(Chaix,"to "related mechanisms (Chaix,"; line 381, "nourseothricin selection(L." to "nourseothricin selection (L.". Line 460, "using 150bp" to "using 150 bp".Line 174 could add the citation to Wang et al., 2019 here as was done in citing the same conclusion on lines 95/96.Line 230 should be "we crossed ras^-1^[bd] to the gcn-5KO strain obtained from the FGSC" or something to that effect. Likewise in the Materials and methods and elsewhere, "bd" should be ras^-1^[bd] or ras^-1^bd. Likewise on line 247 with ada-2 where ras^-1^[bd] was crossed into the knockout strain.More generally it is important that strains always be referred to by their correct and full genotypes unless a clear method of abbreviations is stated early in the main text and invariably applied throughout. This is not trivial, especially with chromatin-modifying proteins. For instance, Belden (PLoS Gen 2011) found a synthetic lethality between ∆chd1 and ras^-1^[bd].

Thanks for the suggestions. The manuscript is revised as suggested. The *ras^-1 bd^* background was revised in the Materials and methods, and stated in the beginning of the main text “we created *cpc-3* and *cpc-1* knockout mutants (See Materials and methods)”.

In the model (Figure 8), formation of the CPC-1 oligomer was not confirmed experimentally, and this could be mentioned.

GCN4/CPC-1 was previously reported to bind DNA in the form of dimers (Hope IA, *et al.*, 1987, EMBO J, PMID: 3678204). We revised our model in Figure 8 and edited the description in the text, “the GCN2 kinase responds to nutrient deprivation, whereas it phosphorylates eIF2α and upregulates the master transcription factors GCN4 or CPC-1, respectively, which binds target DNA as a dimer to activate amino acid biosynthetic genes”.

[Editors' note: further revisions were suggested prior to acceptance, as described below.]

It appears that you have satisfied some, but not all, of the major issues raised about the previous version of the paper. It was recognized that you provided new evidence indicating that knockout of GCN5 eliminates rhythmic FRQ expression with a corresponding effect on FRQ mRNA abundance. While you also provided new ChIP evidence suggesting that CPC-1 occupancy at FRQ is rhythmic; it was provided only to the referees. You performed statistical analysis of rhythmicity using the CircaCompare tool; however, the application of tests and the resulting statistics have not been presented in a transparent or comprehensive manner. It also appears that you incorrectly cited published evidence regarding the rhythmicity of amino acid biosynthetic gene expression. These and a few other issues are fully described below:

Thanks for the comments to our revised manuscript. We have now revised the manuscript to address all raised concerns.

1. The description of the CircaCompare statistical tests of rhythmicity is inadequate and needs to be better explained, providing a brief "layman's" description in the Methods of the meaning of the statistical parameters and how the tests were applied to their data. In addition, the results need to be documented completely for all experiments in which rhythmicity was concluded to exist for any parameter. While the authors added time-series statistics in certain figure legends, they did not indicate the p-values for rhythmicity, and the asterisks in figure panels seem to refer to t-tests rather than to RAIN/CircaCompare statistics. To increase transparency, the results of the CircaCompare statistical tests could be summarized in an additional supplementary table.

We have now revised the manuscript as suggested. Please see our response to the Reviewer #1.

2. Discrepancies remain regarding whether transcription of amino acid (AA) biosynthetic genes is rhythmic as, contrary to the statement in the Response to Reviewers, the publication by Hurley et al. 2014 does not show ser-2 (NCU01439), trp-3 (NCU08409) or arg-1 (NCU02639) to be rhythmic. The authors should note that the original data were re-analyzed using better statistical tools, described in Suppl Dataset S1 of Hurley et al. (2018), and in the re-analyzed data, ser-2(NCU01439) is not rhythmic at the RNA level, although it is rhythmic at the protein level; trp-3(NCU08409) is rhythmic at the RNA level, and arg-1 (NCU02639)- is rhythmic at the RNA level. The authors need to carefully examine the results of Hurley et al. (2018) and provide an accurate summary of the presence or absence of rhythmicity in the expression of all known CPC-1-induced target genes, possibly with an additional supplementary table that lists the key statistical criteria for the assessments. They should also explain to the reviewers why they incorrectly cited the findings of Hurley et al. (2014) in the revised manuscript.

Thanks for the correction and suggestions. We have revised the manuscript as suggested. Please see our response to the Reviewer #1.

3. The new ChIP data for CPC-1 binding at the AA genes should be included as a supplementary figure with the addition of the appropriate explanatory text in Results and Discussion, including the suggestion made in their Response to Reviewers of rhythmic binding of WCC or WCC-controlled transcription factors, at the AA genes and whether any evidence exists in the literature for the latter.

Thanks for the suggestions. We have revised the manuscript as suggested. Please see our response to the Reviewer #1.

4. Clarify whether the authors are interpreting their data to indicate that GCN5/SAGA perform the same functions at FRQ promoter in both non-starved and starved cells and are simply being additionally mobilized by CPC-1 in starved cells, possibly by altering the model in Figure 8.

We have revised the Figure 8 and manuscript as suggested. Please see our response to the Reviewer #2.

Reviewer #1 (Recommendations for the authors):The authors appear to have satisfied some, but not all, of the major issues raised about the previous version of the paper. They have provided new evidence indicating that knock out of GCN5 eliminates rhythmic FRQ expression with a corresponding effect on FRQ mRNA abundance. They also provided new ChIP evidence suggesting that CPC-1 occupancy at FRQ is rhythmic. However, the following issues seem to require additional attention and adequate responses from the authors.1. To support the conclusions of rhythmic FRQ mRNA abundance, WCC complex binding, H3Ac levels, GCN-5 binding, and CPC-1 binding at the FRQ promoter, and of expression levels of amino acid biosynthetic genes in WT cells, they claim to have conducted a statistical analysis using CircaCompare. However, I could not find even a cursory explanation of this tool and the meaning of the different parameters it evaluates in Methods, nor a description of how it was being applied in comparing results for WT and mutants. In addition, it is generally unclear what data have been analyzed with CircaCompare and what the outcomes were. In particular, the legends often state that data were judged to be arhythmic in a cpc mutant without indicating whether it was significantly rhythmic in WT. What is needed is a table that lists clearly all the CircaCompare tests that were conducted and the P-values for all of the parameters that were analyzed, with a final column indicating whether the data are rhythmic or arhythmic and from what results this assessment was made. In short, for each experiment, it is necessary to provide justification using CircaCompare that the data are significantly rhythmic in WT, not only that they are arhythmic in mutants, and to make the CircaCompare tests and results completely transparent, accessible, and understandable to a general audience.

Thanks for the comments and suggestions. We have now revised the manuscript as suggested. We have added in the Methods details of the CircaCompare statistical analysis described in previously published papers (X. Liu et al., 2021; Parsons et al., 2020). The results of the CircaCompare statistical tests were summarized in the Supplementary file 2.

2. The aforementioned shortcoming is particularly notable regarding the expression of amino acid (AA) biosynthetic genes in WT cells. It is unclear whether the CircaCompare analysis actually confirms rhythmic mRNA expression for these genes in WT cells, both for the group of genes they examined by RT-PCR and those interrogated using published RNA-seq data. It is also unclear why they are showing the published RNA-seq results for only 4 genes out of a presumably much larger number of AA biosynthetic genes known to be induced by CPC-1 in starved cells. Did many other AA genes not show rhythmic transcription? It would useful to have a table summarizing the CircaCompare analysis of the RNA-seq data for all known CPC-1 target genes.

Thanks for the suggestions. In the previous manuscript, we only cited the Hurley et al. (2014) paper, because we downloaded and re-analyzed the RNA-seq data in that study. However, the data were re-analyzed using improved statistical methods described in the Supplemental Dataset 1 (https://doi.org/10.17632/r68j3rnxhw.1) of Hurley et al. (2018). We have now cited both papers in our manuscript.

In our study, there were 148 up-regulated and 127 down-regulated genes found in the WT strain after 3-AT treatment but their regulation were abolished in the *cpc-1^KO^* strain (Figure 7D and Figure 7—figure supplement 1A), suggesting that these genes were regulated by CPC-1 under amino acid starvation. As shown in the Supplementary file 1, we summarized the rhythmic expression of 79 up-regulated and 67 down-regulated CPC-1 target genes based on the results of Hurley et al. (2018) (its Supplemental Dataset 1). We also performed RT-qPCR experiments to re-analyze the expression rhythms for some of those genes and found that the rhythmic expression of *his-3* (NCU03139), *trp-3* (NCU08409) and *arg-1* (NCU02639) in the WT was abolished in the *cpc-1^KO^* strain under amino acid starvation, which is consistent with the published RNA-seq analysis results. In addition, we found that *ser-2* (NCU01439) was also rhythmic in the WT but not in the *cpc-1^KO^* strain under amino acid starvation, even though it was not previously shown to be rhythmic in the published RNA-seq analysis study. These results suggest that many CPC-1 activated metabolic genes under amino acid starvation are regulated by circadian clock.

3. This last issue of whether AA biosynthetic gene transcription is truly rhythmic was exacerbated by their new CPC-1 ChIP data (presented only for referees) indicating constitutive CPC-1 occupancy at all four AA biosynthetic genes they tested, in contrast to its apparently rhythmic binding at the FRQ gene. They suggest that the AA biosynthetic genes might exhibit rhythmic binding of WCC, or some other transcriptional activator; however, is there any evidence that WCC binds to these AA genes, or that any other activator besides WCC binds to promoters rhythmically? I remain skeptical of the claim of rhythmic transcription of AA biosynthetic genes that are induced by CPC-1 in starved cells.

Thanks for the concern of this review. Our results and those of Hurley et al. (2018) demonstrated that CPC-1 is required for the rhythmic expression of many CPC-1 targeted metabolic genes. We agree with this reviewer that the exact mechanism for how this regulation is achieved is still unclear since our ChIP experiments showed that the CPC-1 binding at several selected amino acid biosynthetic genes was not rhythmic (Figure 7—figure supplement 1C). There are several possibilities. First, the rhythms of CPC-1 binding at these amino acid biosynthetic genes have very low amplitude and could not be detected by our ChIP assays. Second, the rhythmic transcription of these genes could be controlled by rhythmic binding of WCC or WCC-controlled transcription factors (Hurley et al. 2014). Indeed, the ChIP-seq analysis by Hurley et al. (2014) showed that WC-2 is associated at the promoter of *arg-1* (NCU02639) (https://www.pnas.org/doi/suppl/10.1073/pnas.1418963111/suppl_file/pnas.1418963111.sd11.xlsx). Third, although the binding of CPC-1 is arrhythmic, its activity in activating transcription can still be rhythmic, which could be due to rhythmic posttranslational modification/association of other factors.

We have now revised the manuscript to address this concern. In the results, we described “We performed ChIP experiments to examine whether CPC-1 directly activates the expression of amino acid synthetic genes. As shown in Figure 7—figure supplement 1C, CPC-1 was found to be constitutively enriched at the promoters of *his-3* (NCU03139), *trp-3* (NCU08409) and *ser-2* (NCU01439) genes, and the enrichment was enhanced by 3-AT treatment”. In the discussion, we added that “Although CPC-1 rhythmically binds at the *frq* promoter, ChIP experiments showed that CPC-1 binding at several selected amino acid biosynthetic genes did not appear to be rhythmic (Figure 7—figure supplement 1C). However, our RT-qPCR results (Figure 7F and Figure 7—figure supplement 1) and the previous RNA-seq data showed that many CPC-1 targeted metabolic genes were rhythmic expressed (Hurley et al., 2018). It is possible that the binding of CPC-1 on these promoters are still rhythmic but with low amplitudes. As a result, the limited sensitivity of our ChIP assays failed to detected these rhythms. Alternatively, the rhythmic transcription of these genes might be controlled by rhythmic transcriptional activation activity of CPC-1 rather than its binding. In addition, the rhythmic binding of WCC or WCC-controlled transcription factors (Hurley et al., 2014) might also contribute to their rhythmic transcription.”

Reviewer #2 (Recommendations for the authors):The authors have adequately addressed my comments and concerns from the previous manuscript version, in some cases with additional experiments. I greatly appreciate their effort. I think this study is a nice contribution to the field and will inform on the metabolic regulation of circadian rhythms at the chromatin level. One minor comment that will help clarify this complex model for readers.In the previous version, I commented:"The experiments to examine the involvement of GCN-5 and ADA-2 were performed in normal conditions (no amino acid starvation). Unlike cpc-1 and cpc-3 KO strains, gcn-5 and ada-2 KO strains showed severely disrupted frq rhythms in normal nutrient conditions, suggesting they are normally already required for robust circadian rhythms. If GCN-5 and the SAGA complex are normally involved in regulating H3ac rhythms in the frq loci, how does GCN2 pathway modulates the activity of GCN-5 and SAGA complex in conditions of amino acid starvation? Are the interactions between GCN2/4 with GCN-5 and SAGA complex different in normal vs amino acid starved conditions? "Based on the additional experiments and their text edit, there appears to be no difference in the activity of GCN-5 and SAGA complex in normal vs amino acid-starved conditions. It's doing what it normally does in normal conditions even when it is in amino acid-starved conditions. Its activity is however more necessary in amino acid-starved conditions because the chromatin at frq promoter is constitutively compacted in amino acid-starved conditions. So since CPC-3 and CPC-1 are necessary to activate the GCN-5/ADA-2 complex and since an amino acid-starved condition activates the GCN2 pathway and induced CPC-1 expression, the arrhythmic phenotype in circadian rhythm is more severe in CPC-3 and CPC-1 KO mutants in amino acid starved condition when compared to normal condition.I wonder if the model figure (Figure 8) would be more clear to readers if it includes the normal scenario instead of just having the amino acid-starved conditions. Besides that, I think the manuscript is much approved.

Thanks for the comments and suggestions. It’s correct that the interaction between CPC-1 with ADA-2 and GCN5 is not changed by amino acid starvation. However, the activity of GCN-5 and SAGA complex is more important in the *cpc-3^KO^* and *cpc-1^KO^* mutants compared to the WT strain under amino acid starved conditions. Because the chromatin structure at the *frq* promoter is constitutively compacted by amino acid starvation, CPC-1 protein level is a limiting factor for efficiently recruiting GCN-5 and SAGA complex to loosen the chromatin to drive rhythmic *frq* expression under amino acid starved conditions. As suggested, we have now revised the model in Figure 8 and edited the figure legend and discussion to make it more clear to readers.

[Editors' note: further revisions were suggested prior to acceptance, as described below.]

The manuscript has been improved but there are some remaining issues that need to be addressed, as outlined below:1. In the new Supplementary file S1. The authors have provided lists of 148 cpc-1 up-regulated genes and 127 cpc-1 down-regulated genes, of which 79 up-regulated genes apparently show rhythmic expression and 67 down-regulated genes apparently show rhythmic expression. They have not however provided the statistics of the CircaCompare tests of rhythmicity for the WT strain, which needs to be added for each gene listed in the new file S1. In addition, they should provide new text in the RESULTS explaining that ~53% of both the up- and down-regulated genes exhibit rhythmicity and indicate that this represents a highly significant enrichment of all cpc-1 regulated genes as judged by the hypergeometric distribution test, providing the P-value. The file S1 should also include a notes sheet fully explaining the data provided there.

Thanks for the comments to our revised manuscript. We have now revised the manuscript to address the concerns.

First, it should be noted that the published RNA-seq data were first generated by Hurley et al. in 2014 and were re-analyzed by them using the improved statistical tool eJTK Cycle in 2018. The P-values of eJTK Cycle analysis (from the Supplemental Datasets 1 and 2 of Hurley et al., 2018) has now been added in our revised Supplementary file 1. CircaCompare is used to compare the differences between two rhythmic datasets and CircaSingle (Parsons et al., 2020) is used to analyze the rhythmic parameters of a single group. As suggested by the reviewer, we re-analyzed the rhythmicity of CPC-1 targeted genes (148 up-regulated and 127 down-regulated) using CircaSingle and added the P-values in the revised Supplementary file 1. There were 146 rhythmic genes based on the eJTK Cycle and 132 rhythmic genes based on the CircaSingle. There were 106 overlapping genes between the two sets of data, confirming the results using eJTK Cycle. Thus, we performed further analysis based on the data from eJTK Cycle.

Second, we have now added the description of the rhythmicity of CPC-1 targeted genes in the Results “There were 146/275 (53%) of the CPC-1 up- and down-regulated genes under amino acid starvation exhibiting rhythmicity, indicating a highly significant enrichment of CPC-1 regulated genes as clock-controlled genes (p = 3.341905e-06, hypergeometric distribution test)”. The rhythmic genes and all detected genes of RNA-seq data from Hurley et al. 2018 were added in the revised Supplementary file 1, which was used for the hypergeometric distribution test.

Third, we have now revised the Supplementary file 1 by adding a notes sheet.

2. A notes sheet should also be added to the new Supplementary file S2 in which the results in each sheet should be linked to data in the corresponding figures, and also stipulating the difference between results listed in different sheets for FRQ versus frq (Protein vs. mRNA?). In short, each Supplementary file should be understandable as a stand-alone document.

Thanks for the suggestions. We have now added a notes sheet for Supplementary file 2 and stipulated the information for FRQ protein and *frq* mRNA.

3. There are also some grammatical errors in the revised text that should be corrected.

Thanks for the suggestion. We have carefully checked and improved the English in the revised manuscript.